# Histamine signaling and metabolism identify potential biomarkers and therapies for lymphangioleiomyomatosis

Carmen Herranz[1] , Francesca Mateo[1], Alexandra Baiges[1], Gorka Ruiz de Garibay[1] ,
Alexandra Junza[2,3], Simon R Johnson[4] , Suzanne Miller[4], Nadia García[1], Jordi Capellades[2,3],
Antonio Gómez[5,†] , August Vidal[6,7] , Luis Palomero[1], Roderic Espín[1], Ana I Extremera[1],
Eline Blommaert[1], Eva Revilla-López[8], Berta Saez[8], Susana Gómez-Ollés[8], Julio Ancochea[9],
Claudia Valenzuela[9], Tamara Alonso[9], Piedad Ussetti[10], Rosalía Laporta[10], Antoni Xaubet[11,‡],
José A Rodríguez-Portal[12,13], Ana Montes-Worboys[13,14], Carlos Machahua[13,14] , Jaume Bordas[13,14] ,
Javier A Menendez[1], Josep M Cruzado[15,16], Roser Guiteras[15,16], Christophe Bontoux[17],
Concettina La Motta[18] , Aleix Noguera-Castells[19], Mario Mancino[19], Enrique Lastra[20] ,
Raúl Rigo-Bonnin[21] , Jose C Perales[22], Francesc Viñals[1,22] , Alvaro Lahiguera[1], Xiaohu Zhang[23],
Daniel Cuadras[24] , Coline H M van Moorsel[25], Joanne J van der Vis[25], Marian J R Quanjel[25],
Harilaos Filippakis[26] , Razq Hakem[27], Chiara Gorrini[28], Marc Ferrer[23], Aslihan Ugun-Klusek[29] ,
Ellen Billett[29], Elżbieta Radzikowska[30] , Álvaro Casanova[31], María Molina-Molina[13,14],
Antonio Roman[8] , Oscar Yanes[2,3] & Miquel A Pujana[1,*]

1   ProCURE, Catalan Institute of Oncology, Oncobell, Bellvitge Institute for Biomedical Research (IDIBELL), L'Hospitalet del Llobregat, Barcelona, Spain
2   Department of Electronic Engineering, Institute of Health Research Pere Virgili (IIPSV), University Rovira i Virgili, Tarragona, Spain
3   Biomedical Research Network Centre in Diabetes and Associated Metabolic Diseases (CIBERDEM), Instituto de Salud Carlos III, Madrid, Spain
4   National Centre for Lymphangioleiomyomatosis, Nottingham University Hospitals NHS Trust, Nottinghamshire, Division of Respiratory Medicine, University of Nottingham, Nottingham, UK
5   Centre for Genomic Regulation, Barcelona Institute of Science and Technology, Barcelona, Spain
6   Department of Pathology, University Hospital of Bellvitge, Oncobell, IDIBELL, L'Hospitalet del Llobregat, Barcelona, Spain
7   CIBER on Cancer (CIBERONC), Instituto de Salud Carlos III, Madrid, Spain
8   Lung Transplant Unit, Pneumology Service, Lymphangioleiomyomatosis Clinic, Vall d'Hebron University Hospital, Barcelona, Spain
9   Pneumology Service, La Princesa Research Institute, University Hospital La Princesa, Madrid, Spain
10  Pneumology Service, University Hospital Clínica Puerta del Hierro, Majadahonda, Madrid, Spain
11  Pneumology Service, Hospital Clínic de Barcelona, Barcelona, Spain
12  Medical-Surgical Unit of Respiratory Diseases, Institute of Biomedicine of Seville (IBiS), University Hospital Virgen del Rocío, Seville, Spain
13  Biomedical Research Network Centre in Respiratory Diseases (CIBERES), Instituto de Salud Carlos III, Madrid, Spain
14  Interstitial Lung Disease Unit, Department of Respiratory Medicine, University Hospital of Bellvitge, IDIBELL, L'Hospitalet del Llobregat, Barcelona, Spain
15  Experimental Nephrology, Department of Clinical Sciences, University of Barcelona, Barcelona, Spain
16  Department of Nephrology, University Hospital of Bellvitge, IDIBELL, L'Hospitalet del Llobregat, Barcelona, Spain
17  Department of Pathology, University Hospital Pitié-Salpêtrière, Faculty of Medicine, University of Sorbonne, Paris, France
18  Department of Pharmacy, University of Pisa, Pisa, Italy
19  Biomedical Research Institute "August Pi i Sunyer" (IDIBAPS), Department of Medicine, University of Barcelona, Barcelona, Spain
20  Genetic Counseling Unit, Department of Medical Oncology, University Hospital of Burgos, Burgos, Spain
21  Clinical Laboratory, University Hospital of Bellvitge, IDIBELL, L'Hospitalet de Llobregat, Barcelona, Spain
22  Department of Physiological Science II, University of Barcelona, Barcelona, Spain
23  National Center for Advancing Translational Sciences (NCATS), National Institute of Health (NIH), Bethesda, MD, USA
24  Statistics Department, Foundation Sant Joan de Déu, Esplugues, Spain
25  Interstitial Lung Disease (ILD) Center of Excellence, St. Antonius Hospital, Nieuwegein, The Netherlands
26  Pulmonary and Critical Care Medicine, Department of Medicine, Brigham and Women's Hospital, Harvard Medical School, Boston, MA, USA
27  Princess Margaret Cancer Centre, University Health Network, Department of Medical Biophysics, University of Toronto, Toronto, Ontario, Canada
28  Princess Margaret Hospital, The Campbell Family Institute for Breast Cancer Research, Ontario Cancer Institute, University Health Network, Toronto, ON, Canada
29  Centre for Health, Ageing and Understanding Disease (CHAUD), School of Science and Technology, Nottingham Trent University, Nottingham, UK
30  Department of Lung Diseases III, National Tuberculosis and Lung Disease Research Institute, Warsaw, Poland
31  Pneumology Service, University Hospital of Henares, University Francisco de Vitoria, Coslada, Madrid, Spain
   *Corresponding author. Tel: +34 932607952; E-mail: mapujana@iconcologia.net
   †Present address: Rheumatology Department and Rheumatology Research Group, Vall d'Hebron Hospital Research Institute (VHIR), Barcelona, Spain
   ‡Deceased

## Abstract

Inhibition of mTOR is the standard of care for lymphangioleiomyomatosis (LAM). However, this therapy has variable tolerability and some patients show progressive decline of lung function despite treatment. LAM diagnosis and monitoring can also be challenging due to the heterogeneity of symptoms and insufficiency of non-invasive tests. Here, we propose monoamine-derived biomarkers that provide preclinical evidence for novel therapeutic approaches. The major histamine-derived metabolite methylimidazoleacetic acid (MIAA) is relatively more abundant in LAM plasma, and MIAA values are independent of VEGF-D. Higher levels of histamine are associated with poorer lung function and greater disease burden. Molecular and cellular analyses, and metabolic profiling confirmed active histamine signaling and metabolism. LAM tumorigenesis is reduced using approved drugs targeting monoamine oxidases A/B (clorgyline and rasagiline) or histamine H1 receptor (loratadine), and loratadine synergizes with rapamycin. Depletion of *Maoa* or *Hrh1* expression, and administration of an L-histidine analog, or a low L-histidine diet, also reduce LAM tumorigenesis. These findings extend our knowledge of LAM biology and suggest possible ways of improving disease management.

**Keywords** biomarker; histamine; lymphangioleiomyomatosis; mTOR; therapy
**Subject Categories** Cancer; Metabolism; Respiratory System

## Introduction

Lymphangioleiomyomatosis (LAM) is a rare low-grade neoplasm that affects almost exclusively women and is characterized by cystic lung destruction, which can lead to fatal respiratory failure in severe cases (Johnson *et al*, 2010; McCormack *et al*, 2016). Extrapulmonary manifestations of LAM include chylous abdominal ascites, lymphadenopathy, lymphangioleiomyomas and, most commonly, renal angiomyolipomas (AML) (Johnson *et al*, 2010; McCormack *et al*, 2016). Lung destruction is caused by abnormal proliferation of smooth muscle-like and epithelioid diseased cells with predicted metastatic potential (Crooks *et al*, 2004; Yu *et al*, 2009; Terasaki *et al*, 2010; Henske & McCormack, 2012; Krymskaya & McCormack, 2017). LAM can occur sporadically (S-LAM) or in presence of tuberous sclerosis complex (TSC-LAM), an autosomal-dominant multisystem disorder caused by germline or mosaic loss-of-function mutations in the tumor suppressor genes *TSC1* and *TSC2* (Crino *et al*, 2006; Giannikou *et al*, 2016). Most LAM cases are sporadic and somatic inactivation of *TSC2* in an unknown cell type(s) is expected to be the central genetic alteration necessary for disease development (Carsillo *et al*, 2000). Germline or somatic mutations in *TSC1/TSC2* cause abnormal activation of the mechanistic target of rapamycin (mTOR) (Goncharova *et al*, 2002; Saxton & Sabatini, 2017), and this is the basis of the standard of care use of mTOR allosteric inhibitors for LAM (McCormack *et al*, 2011). Through its central role in metabolism, mTOR activity is abnormally enhanced in many cancer types, and generally linked to stem cell-like features, which are also present in LAM cells (Krymskaya, 2008; Ruiz de

Garibay *et al*, 2015; Julian *et al*, 2017; Pacheco-Rodríguez *et al*, 2019).

The diagnosis of LAM is challenging because of the variability of its clinical symptoms and its relative similarity to other interstitial lung pathologies. Reliable diagnosis is generally achieved when patients present with defined lung cysts and other clear manifestations, particularly AML (Johnson *et al*, 2010; McCormack *et al*, 2016). In this context, the Multicenter International Lymphangioleiomyomatosis Efficacy and Safety of Sirolimus (MILES) trial (McCormack *et al*, 2011) highlighted the clinical value of measuring basal serum levels of VEGF-D as a disease biomarker (Young *et al*, 2013). This biomarker has been clinically implemented in several centers and can circumvent the need for a lung biopsy for diagnosis in some cases. However, a relatively low VEGF-D level does not rule out a diagnosis of LAM and, therefore, patients who do not show the most typical symptoms may still require a lung biopsy (Johnson *et al*, 2010; McCormack *et al*, 2016). In addition, basal levels of VEGF-D may fluctuate due to unknown individual factors, and its overabundance may be primarily linked to specific pathological features, mainly lymphatic abnormalities (Taveira-DaSilva *et al*, 2018). Therefore, complementary biomarkers may help consolidate protocols for non-invasive disease diagnosis and monitoring, while providing further insight into LAM biology.

The MILES trial was a remarkable success in that it demonstrated that the use of rapamycin, an allosteric mTOR inhibitor also known as sirolimus, is beneficial for treating LAM (McCormack *et al*, 2011). This therapy significantly stabilizes pulmonary function and reduces AML size in most cases, establishing the current standard of care for the disease. However, this treatment does not fully eradicate LAM cells and, although mTOR inhibition has the drawback of immunosuppression, discontinuation of treatment generally leads to disease reversion (McCormack *et al*, 2011). Rapamycin has considerably improved outcomes for women with LAM; however, a significant minority of patients continues to lose lung function, albeit at a lower rate, while taking rapamycin (Young *et al*, 2013; Bee *et al*, 2015, 2018). Therefore, much effort is being directed toward identifying therapies that, as single agents or in combination with a reduced rapamycin dose, can improve disease care by providing complementary therapeutic options for some patients. Here, using a cross-disease approach, we propose LAM plasma biomarkers that, in turn, have led to the preclinical demonstration of beneficial therapeutic approaches in monotherapy or in combination with rapamycin. The novel therapies are based on drugs approved for other human conditions. A potential biomarker derives from histamine degradation and is independent of VEGF-D. The proposed biomarkers and therapies are connected by novel insight into active histamine metabolism and signaling in LAM.

## Results

### Inference from breast cancer metastatic profiles leads to the identification of novel LAM plasma metabolite biomarkers

Loss of *TSC2* expression and, therefore, abnormally enhanced mTOR activity, have been associated with lung-metastatic potential in breast cancer (Jiang *et al*, 2005; Nasr *et al*, 2013; Ruiz de Garibay *et al*, 2015; Mateo *et al*, 2017). Based on this, we analyzed gene

expression profiles of breast cancer in search of predictions of active LAM metabolic pathways and, as a consequence, potential plasma biomarkers (Fig 1A). Identification of LAM biomarkers could uncover therapeutic opportunities for managing the disease (Fig 1A). By analyzing a gene expression dataset of breast tumors that metastasize to the lung (Minn *et al*, 2005), we identified genes negatively correlated with *TSC2* (Pearson's correlation coefficient (PCC) < 0, and one-sided $P < 0.05$). Among these genes, those coding for enzymes were selected ($n = 30$; Appendix Table S1A). Examination of this gene set predicted several active metabolic pathways, some of which, such as the mitochondrial β-oxidation and pentose phosphate pathways, had previously been linked to LAM biology (Fig 1B) (Düvel *et al*, 2010; Yecies & Manning, 2011; Sun *et al*, 2014; Li *et al*, 2016).

Building on the above predictions, absolute quantification assays using liquid chromatography (LC) combined with mass spectrometry (MS; LC-MS/MS) were performed for seven metabolites (Appendix Table S1B) in plasma samples obtained from 22 LAM patients and three healthy pre-menopausal women. Four metabolites showed a trend toward overabundance in LAM samples (Mann–Whitney test; $P < 0.05$) and were subsequently analyzed in plasma from an additional set of 30 LAM patients (Appendix Table S2), 12 healthy age-matched female controls, as well as samples from women with LAM-related pulmonary diseases, the latter group comprising 10 individuals diagnosed with Langerhans cell histiocytosis, 10 with Sjögren syndrome, seven with emphysema, and three with systemic lupus erythematosus. Metabolite profiling identified a significantly greater abundance of 1-methylimidazole-4-acetic acid (MIAA) in LAM plasma than in any other group (Fig 1C). The lowest 10% of MIAA measures in LAM patients were obtained from 75% of all the patients with related pulmonary diseases. The level of MIAA was not correlated with that of VEGF-D in LAM plasma (Fig 1D), but was significantly more abundant in patients who did not receive rapamycin treatment (Fig 1E). It was not possible to evaluate the associations of metabolites with pulmonary function in this cohort.

Next, stepwise modeling indicated that MIAA alone or in combination with other metabolites (subsequent section) did not improve the predictive accuracy of VEGF-D alone in comparisons of LAM with healthy controls (Fig 1F, left panel). However, the increase in the area under the receiver operating characteristic (ROC) curve from 82% to 91% (Fig 1F, right panel) indicated that the combination of VEGF-D and MIAA improved the accuracy in this LAM cohort compared with patients with related pulmonary diseases.

The three additional metabolites tested in the validation assays (homovanillic acid (HVA), 4-hydroxyphenylacetic acid (4-HPA), and 3-methoxy-4-hydroxymandelic acid (VMA)) also showed significant differences among the sample groups, although to a lesser extent than that observed with MIAA. All three were more abundant in LAM than in healthy plasma, and HVA was also more abundant in LAM plasma than in related pulmonary diseases (Fig 1G and Appendix Fig S1). All identified metabolites were derived from the catabolism of monoamines mediated by monoamine oxidases A/B (MAO-A/B) and aldehyde dehydrogenases (ALDHs) in the mitochondria (Fig 1H), which may have increased the reactive oxygen species (ROS) (Ugun-Klusek *et al*, 2019).

### Clinical associations of plasma histamine and MIAA in an independent LAM cohort

1-Methylimidazole-4-acetic acid is the major metabolite of histamine detected in body fluids. Histamine metabolism (Fig 1H) is a normal process that occurs in most tissues (Schwelberger *et al*, 2013), but is altered in a range of neoplasms, including breast cancer, where it may promote cancer progression (Garcia-Caballero *et al*, 1994; Reynolds *et al*, 1998; Medina *et al*, 2006; von Mach-Szczypiński *et al*, 2009; Fritz *et al*, 2020). To evaluate the potential of MIAA and of histamine as LAM biomarkers, plasma samples from 20 healthy women and 20 patients not treated with rapamycin and collected in an independent cohort (Appendix Table S3) were analyzed for both metabolites using LC-MS/MS. MIAA was confirmed to be more

**Figure 1. Identification of LAM plasma biomarkers.**

A  Strategy for identifying metabolic plasma biomarkers and, subsequently, therapeutic opportunities for LAM based on the analysis of lung-metastatic breast cancer gene expression data (Minn *et al*, 2005).

B  Significantly enriched metabolic pathways among the 30 enzymes identified in the previous analysis of *TSC2* expression correlations.

C  Overabundance of MIAA in LAM plasma. Top panel shows each control and patient group (number (*n*) of samples are indicated); bottom panel shows aggregation of related pulmonary diseases. The asterisks indicate significant differences based on two-sided Mann–Whitney tests (*$P < 0.05$, **$P < 0.01$, ***$P < 0.001$, and ****$P < 0.0001$; top panel, control-LAM $P = 0.048$, LAM-Langerhans $P = 1 \times 10^{-4}$, LAM-Sjögren $P = 1 \times 10^{-4}$, LAM-lupus $P = 1 \times 10^{-3}$, LAM-emphysema $P = 0.041$; bottom panel, LAM-Langerhans $P = 7 \times 10^{-6}$ and control-other $P = 0.031$). Average values are indicated with lilac-colored lines. Comparison of the three groups: Kruskal–Wallis test $P = 6 \times 10^{-4}$ (top panel) and $P = 3 \times 10^{-5}$ (bottom panel).

D  Lack of correlation between MIAA and VEGF-D plasma measures (Spearman's $r_s$ n.s., not significant).

E  Overabundance of MIAA in LAM patients not treated with rapamycin relative to rapamycin-treated (number (*n*) of samples are indicated). The asterisk indicates significant difference based on two-sided Mann–Whitney test ($P = 0.043$). Average values are indicated with lilac-colored lines.

F  Receiver operating characteristic (ROC) curves and the corresponding area under the curve (AUC) and Akaike information criterion (AIC) values for the analyzed metabolites plus VEGF-D, comparing LAM and healthy (left) or related pulmonary disease (right) plasma.

G  Plasma levels of three other metabolic products derived from monoamine metabolism and showing significant differences from controls and/or between LAM and related pulmonary diseases (individual samples indicated in panel C). The asterisks indicate significant differences based on two-sided Mann–Whitney tests (*$P < 0.05$, **$P < 0.01$, and ****$P < 0.0001$; 4-HPA, control-LAM $P = 0.024$ and LAM-other $P = 0.089$; HVA, control-LAM $P = 0.008$ and LAM-other $P = 0.013$; and VMA, control-LAM $P = 1 \times 10^{-4}$ and LAM-other $P = 0.10$). Average values are indicated with lilac-colored lines. Detailed results by patient group are shown in Appendix Fig S1. Comparison of the three groups: Kruskal–Wallis test; 4-HPA $P = 0.095$; HVA $P = 0.013$; and VMA $P = 0.005$.

H  Left panel, concise representation of the mitochondrial reactions mediated by MAOs and ALDHs from a monoamine (R-NH$_2$), to an aldehyde (R-CHO), to the four identified metabolites (R-COOH). The main monoamine origin of each metabolite is indicated in parentheses. Right panel, detail of the histamine metabolism pathway, with enzymes or enzymatic activities marked in red: ALDH, diamine oxidase (DAO), histidine decarboxylase (HDC), histamine N-methyltransferase (HMT), imidazole acetic acid-phosphoribosyl transferase (IPRT), and MAO-B.

Source data are available online for this figure.

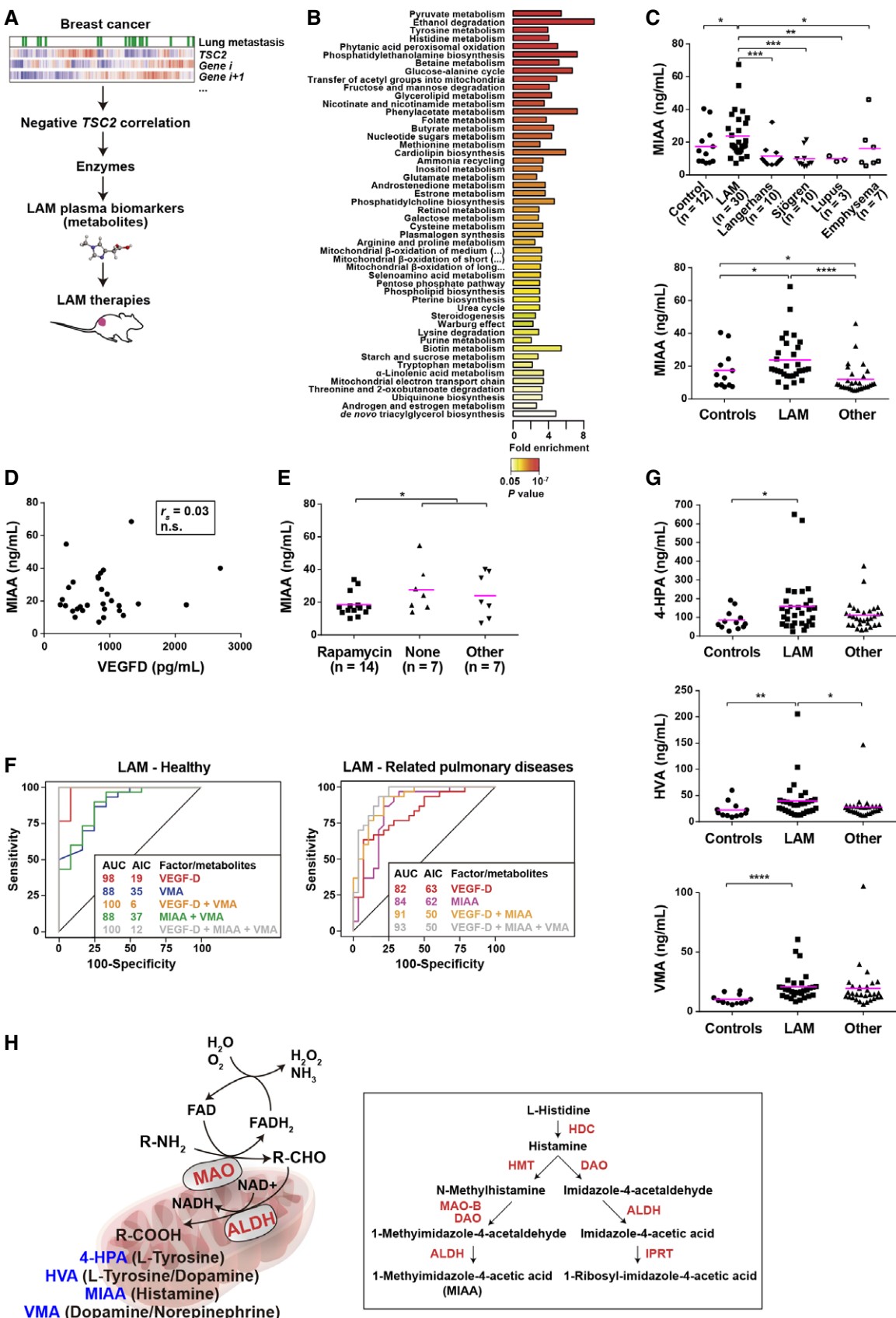

**Figure 1.**

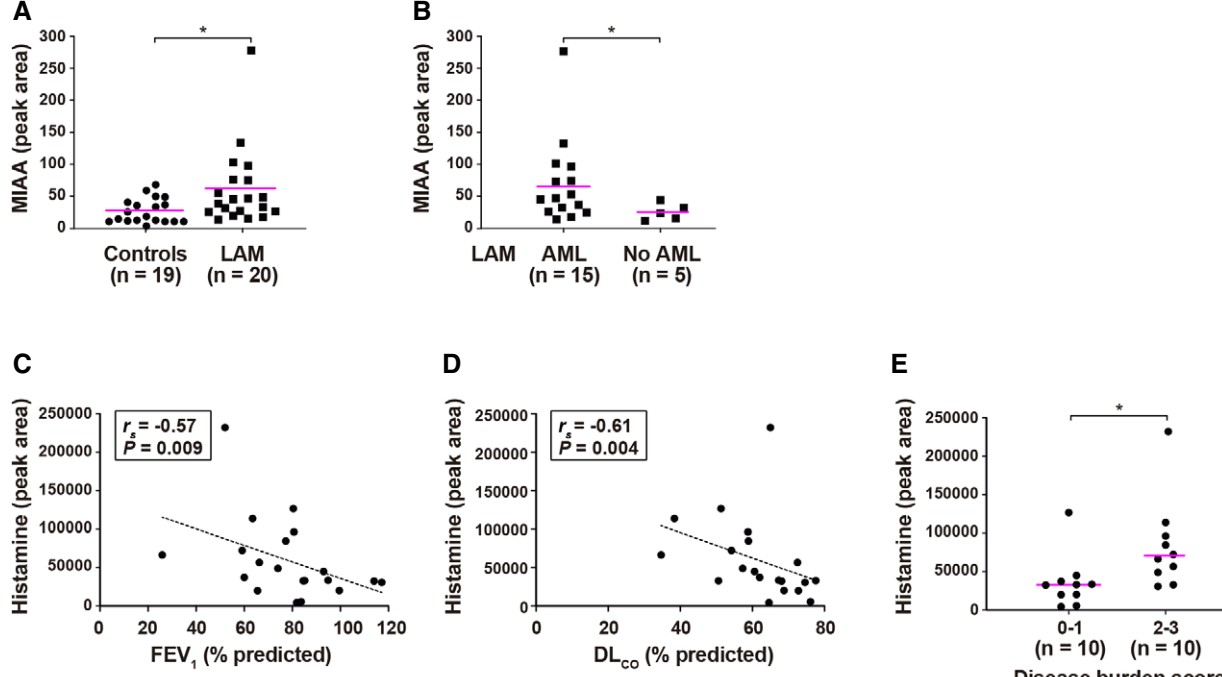

**Figure 2. Validation of MIAA and identification of associations between histamine and LAM clinical features.**

A  Overabundance of MIAA in LAM plasma (UK cohort not treated with rapamycin) relative to healthy women. The number (*n*) of individuals in each group is indicated; one of 20 healthy controls did not pass the quality controls for LC-MS/MS. Asterisk indicates significance using two-sided Mann–Whitney test (*P* = 0.018). Average values are indicated with lilac-colored lines.

B  Overabundance of MIAA in plasma of LAM patients with AMLs. Asterisk indicates significance using two-sided Mann–Whitney test (*P* = 0.042). The number (*n*) of individuals in each group is indicated. Average values are indicated with lilac-colored lines.

C  Negative correlation (*r*$_s$ and *P* are indicated) between histamine levels in LAM plasma and patient FEV$_1$ (% predicted), at the time of blood sample collection.

D  Similar negative correlation (*r*$_s$ and *P* are indicated) between histamine and DL$_{CO}$.

E  Overabundance of MIAA in plasma of LAM patients with higher disease burden. Asterisk indicates significance using two-sided Mann–Whitney test (*P* = 0.015). The number (*n*) of individuals in each group is indicated. Average values are indicated with lilac-colored lines.

Source data are available online for this figure.

---

abundant in LAM patients than in controls (Fig 2A). In addition, MIAA was relatively more abundant in LAM with AML (Fig 2B). Histamine levels did not differ between LAM and controls, but were significantly negatively correlated with measures of forced expiratory volume in the first second (FEV$_1$) and with the diffusing capacity of the lungs for carbon monoxide (DL$_{CO}$) in LAM (Fig 2C and D). Higher histamine levels in LAM plasma were also associated with greater disease burden scores (Fig 2E). Histamine did not reach significant correlation with VEGF-D (Spearman's correlation coefficient ($r_s$) = 0.42, $P$ = 0.065), but as expected, was significantly negatively correlated with MIAA ($r_s$ = −0.47, $P$ = 0.038). MIAA was not significantly correlated with FEV$_1$ or DL$_{CO}$, but the positive trends were in the expected direction: $r_s$ = 0.28, $P$ = 0.24; and $r_s$ = 0.25, $P$ = 0.30, respectively. In addition, as observed in the previous LAM cohort, MIAA was not correlated with VEGF-D: $r_s$ = −0.08, $P$ = 0.75.

**Increased ALDH and MAO expression, and MAO activity in a LAM cell model**

The identified monoamine-derived metabolites may have originated from LAM lung lesions and/or the surrounding affected tissue. To assess a source from LAM cells, we first analyzed the expression of key genes and proteins. Our previous investigation showed frequent ALDH1 positivity in LAM lung lesions (Ruiz de Garibay *et al*, 2015). In the current study, we detected positivity of MAO-A and MAO-B in all lung lesions of seven LAM patients analyzed (Fig 3A, top panels). In lung tissue from healthy individuals, MAOs were expressed in the alveolar epithelium and the luminal layer of the bronchioles (Fig 3A, bottom panels). Co-staining with the LAM marker α-smooth muscle actin (αSMA) revealed co-localization with both MAO-A and MAO-B positivity in LAM lesions, with MAO-B being confined to pathological areas (Fig 3B). Control immunochemistry results of αSMA are shown in Appendix Fig S2. Analysis of gene expression data from microdissected LAM lung nodules (Pacheco-Rodriguez *et al*, 2009) provided further evidence of overexpression of *MAOA/B*—and of other genes in the monoamine-histamine metabolism pathway—in these lesions relative to pulmonary-artery smooth muscle cells (Appendix Fig S3). In addition, a recent landmark study of transcriptome profiles of single cells from LAM lung tissue has defined a LAM$^{core}$ signature that includes *MAOB* and *ALDH* genes (Guo *et al*, 2020).

The expression of the *Aldh* and *Mao* genes and/or gene products was then evaluated in a LAM cell model and its corresponding

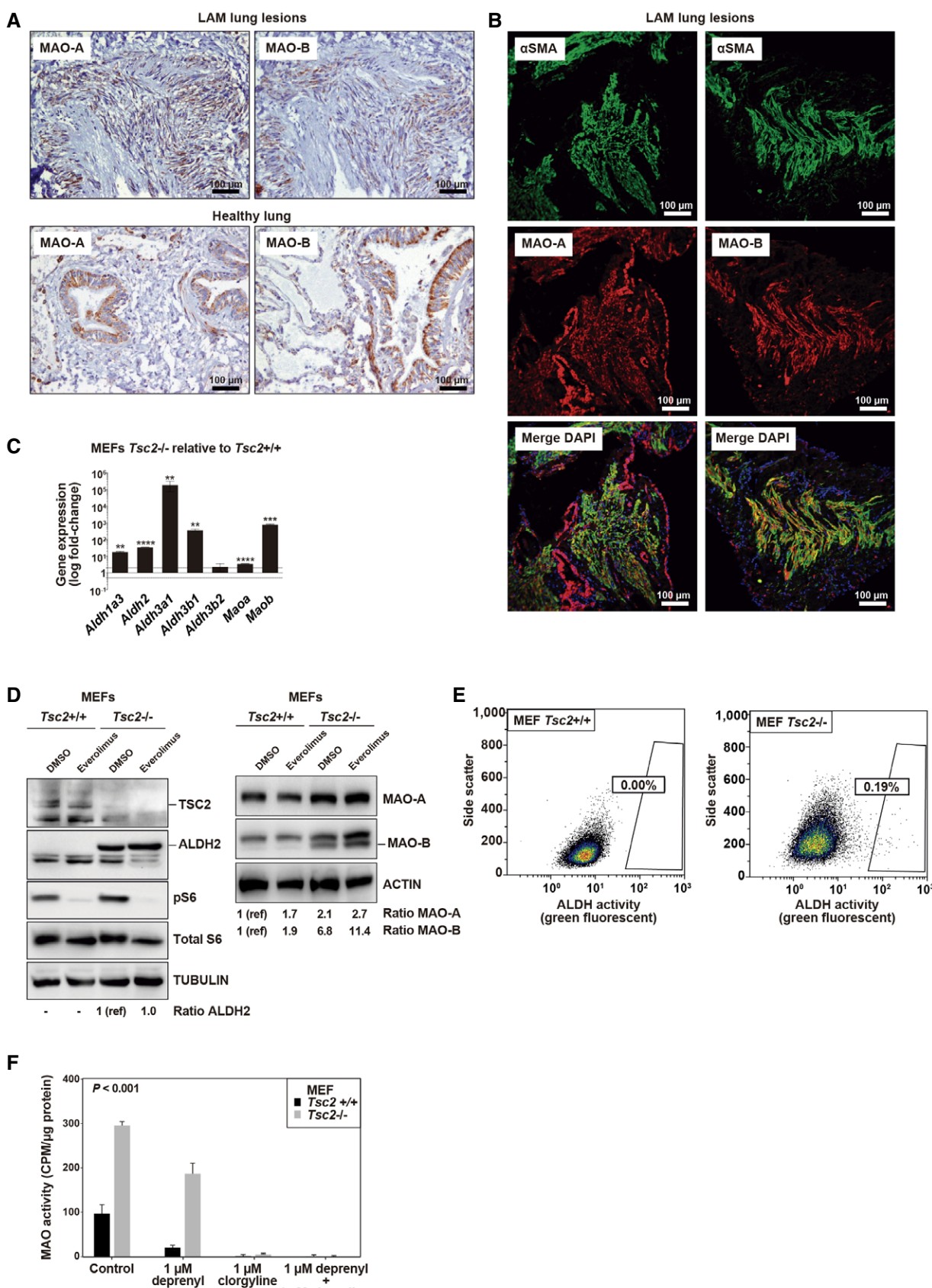

**Figure 3.**

**Figure 3.  Expression and function of ALDHs and MAOs in LAM tissue and cell model.**

A   Representative images of immunohistochemical positivity of MAO-A/B in LAM lung lesions (top panels, two patients) and lung tissue from healthy (non-LAM) individuals (bottom panels). In total, seven LAM patients and three healthy controls were analyzed. Brown-stained cells, counter-stained with hematoxylin, are considered positive. Scale bars are shown.

B   Representative images of immunofluorescence detection of MAO-A/B and αSMA in LAM lung lesions, nuclei stained with DAPI (merged).

C   Graph showing gene expression differences ($log_{10}$-fold changes) of defined genes (X-axis) in *Tsc2*-deficient relative to wild-type MEFs grown in DMEM 10% FBS. The asterisks indicate significant differences with two-sided *t*-test (**$P < 0.01$, ***$P < 0.001$, and ****$P < 0.0001$; *Aldh1a3* $P = 0.006$, *Aldh2* $P = 2 \times 10^{-5}$, *Aldh3a1* $P = 0.004$; *Aldh3b1* $P = 0.006$; *Aldh3b2*, $P = 0.17$, *Maoa* $P = 7 \times 10^{-5}$, and *Maob* $P = 7 \times 10^{-4}$; replicates/condition $n = 3$, assays $n = 4$). The bars indicate mean ± SD. Dotted horizontal lines indicate 2-fold (top) and 0.5-fold (bottom).

D   Western blot results (independent experiments $n = 4$) from *Tsc2*-deficient and wild-type MEFs exposed to DMSO or treated with 20 nM everolimus for 16 h in DMEM 10% FBS. Loading controls are shown. The inferred protein expression level is indicated by the ratio between the corresponding signal and loading control, standardized to the basal setting (noted as 1 (reference (ref))).

E   Flow cytometry results (independent experiments $n = 3$) showing cell percentages for ALDEFLUOR-positive MEFs grown in DMEM 10% FBS.

F   Quantified MAO basal activity (Y-axis) in MEFs as depicted in the inset (grown in DMEM 10% FBS; replicates/condition $n = 3$ and independent experiments $n = 2$). The inhibitors (depicted on the X-axis) were added to cell extracts prior to activity assay to assess the contribution of each MAO isoform. The significant difference corresponds to two-way ANOVA test ($P = 8 \times 10^{-4}$). CPM: counts per minute. The bars indicate mean ± standard error of the mean (SEM).

Source data are available online for this figure.

control counterpart, i.e., *Tsc2*-deficient or *Tsc2*-wild-type and *Trp53*-deficient mouse embryonic fibroblasts (MEFs). Relatively higher levels of most *Aldh* isoforms and of *Maoa* and *Maob* were detected in *Tsc2*-deficient MEFs (Fig 3C). Western blot assays confirmed the increase in protein expression of ALDH2 and MAO isoforms, and the levels of MAO-B appeared to be further increased by exposure to the rapalog everolimus (Fig 3D).

*Tsc2*-deficient MEFs cells showed a slight, but consistent emergence of ALDH-positive cells as measured by ALDEFLUOR-based assays: mean *Tsc2* wild-type percentage = 0.02 (standard deviation (SD) = 0.03); mean *Tsc2*-deficient percentage = 0.31 (SD = 0.19) (Fig 3E). Notably, an independent study showed that circulating human LAM cell populations have comparatively higher ALDH activity (Pacheco-Rodríguez *et al*, 2019). Subsequently, *in vitro* enzymatic activity assays revealed greater basal MAO activity in *Tsc2*-deficient MEFs than in control counterparts (Fig 3F). Cells exposed to 1 μM clorgyline (approved MAO-A inhibitor) for 48 h showed a reduction of this activity to zero; separately, exposure to 1 μM deprenyl (approved MAO-B inhibitor) for the same period reduced activity by approximately 35% (Fig 3F). Therefore, these effects appeared to be consistent with the basal protein expression of the enzymes (Fig 3D), suggesting that both isoforms are active in *Tsc2*-deficient cells. However, while the 1-μM drug concentrations used in these assays were based on previous evidence from cells and tissue and were used with the intention of achieving complete enzymatic inhibition (Ugun-Klusek *et al*, 2019), we could not rule out the possibility that MAO cross-inhibition had occurred in the assays. Collectively, the expression and function of ALDH and MAO appear to be stronger in LAM-related cells, and such overactivation may account for the enhanced detection of monoamine-derived metabolites in the plasma of LAM patients.

**Molecular features associated with increased monoamine metabolism**

Alteration or modification of mitochondrial biology has been recognized in *Tsc1/Tsc2*-deficient cells (Goto *et al*, 2011; Ebrahimi-Fakhari *et al*, 2016; Obraztsova *et al*, 2020; Condon *et al*, 2021). Although the differences observed between *Tsc2*-deficient and wild-type MEFs might be due to altered mitochondrial content, analyses of *Vdac1*, which codes for the most abundant protein in the

mitochondrial outer membrane, failed to reveal expression differences between the two MEF lines (Fig 4A). Nevertheless, differences in mitochondrial morphology and homeostasis may still be present (Ebrahimi-Fakhari *et al*, 2016; Abdelwahab *et al*, 2018). Next, since the greater activity of ALDH and MAO may be linked to higher levels of mitochondrial ROS, these species were measured using MitoSOX red-based assays. The ROS signal of the *Tsc2*-deficient MEFs was more positive and intense than that of the wild-type counterpart (Fig 4B). The mitochondrial membrane potential of the *Tsc2*-deficient cells, as measured by MitoTracker red-based assays, was also higher (Fig 4C). Likewise, these MEFs exhibited higher values of total basal respiration (Fig 4D). These findings are consistent with those of pivotal studies demonstrating high levels of ROS in *TSC2/Tsc2*-deficient cells and, consequently, greater sensitivity to enhanced oxidative stress (Finlay *et al*, 2005; Medvetz *et al*, 2015; Filipczak *et al*, 2016; Li *et al*, 2016; Lam *et al*, 2017).

Since the MAO enzymes specifically produce hydrogen peroxide by oxidation of monoamines, this ROS form was subsequently measured in MEF cultures. The *Tsc2*-deficient cells showed significantly higher levels of hydrogen peroxide than did their wild-type counterparts (Fig 4E). This compound is decomposed into water and oxygen by the action of various enzymes, of which catalase is particularly important, and this protein was also overexpressed in *Tsc2*-deficient MEFs (Fig 4F). Consistent with high levels of ROS, all analyzed LAM lung tissue (7/7 patients) were found to be positive for acrolein, an aldehyde that is generated endogenously through oxidation reactions (Fig 4G and Appendix Fig S2).

**Histamine metabolism and HRH1 signaling in LAM**

While the metabolism of several monoamines appears to be enhanced in LAM, the detection of MIAA could indicate a major role for histamine. To evaluate this, we used LC-MS/MS to assay the levels of MIAA and histamine in *Tsc2*-deficient MEFs and their control counterparts. The results of five replicates of cell cultures in complete media (10% fetal bovine serum (FBS)) showed significantly lower content of histamine in *Tsc2*-deficient MEFs (Fig 5A). Of note, global metabolic profiling also identified lower levels of histamine in *Tsc2*-deficient MEFs (Düvel *et al*, 2010).

Next, we performed stable isotope labeling assays using [13C$_6$,15N$_3$]-labeled L-histidine provided to MEF cultures over

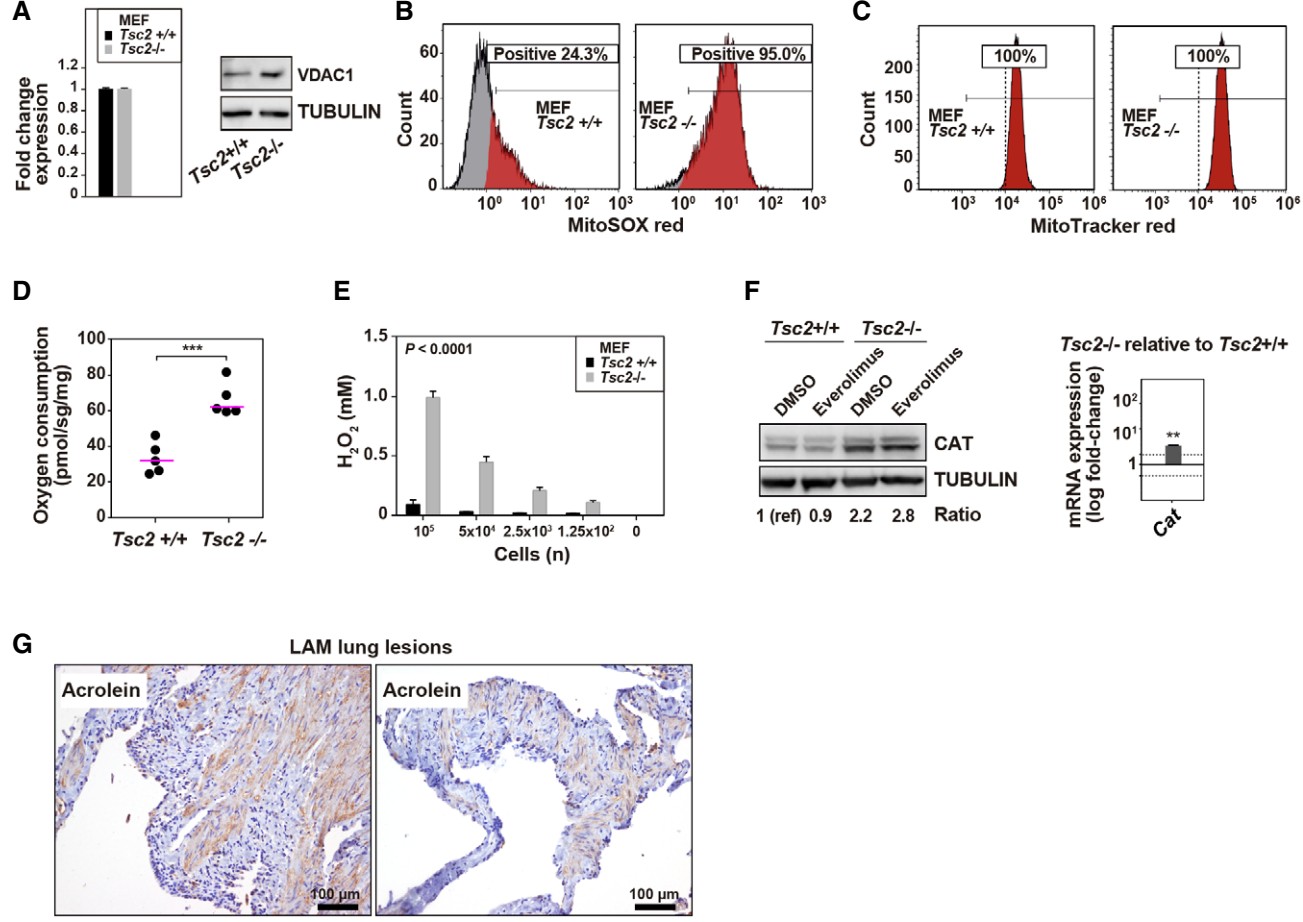

**Figure 4. Enhanced mitochondrial activity and H$_2$O$_2$ in the LAM cell model and consequences for LAM lesions.**

A No differences of *Vdac1*/VDAC1 expression between MEF cell lines (replicates/condition $n = 3$ and independent experiments $n = 2$). The graph shows fold change expression (mean $\pm$ SEM) in *Tsc2*-deficient relative to wild-type MEFs.

B Flow cytometry results (independent experiments $n = 3$) showing a higher percentage of MitoSOX red-positive cells in *Tsc2*-deficient MEFs.

C Flow cytometry results (independent experiments $n = 3$) showing higher intensity (X-axis, single channel intensity FL3 670 nm) of MitoTracker red-positive cells in *Tsc2*-deficient MEFs.

D Higher basal cell respiration (as measured by oxygen consumption, Y-axis) in *Tsc2*-deficient MEFs. The asterisks indicate significant difference with two-sided *t*-test ($P = 5 \times 10^{-4}$; replicates/condition $n = 5$, and independent experiments $n = 3$).

E Higher hydroxide peroxide levels in *Tsc2*-deficient MEFs across different numbers of seeded cells (X-axis). Significance corresponds to two-way ANOVA test ($P = 1 \times 10^{-4}$; replicates/condition $n = 2$, independent experiments $n = 2$). The bars indicate mean $\pm$ SD.

F Left panel, Western blot results showing overexpression of catalase (CAT) in *Tsc2*-deficient MEFs, and unaffected by exposure to everolimus (independent experiments $n = 2$). The CAT expression level is indicated by the ratio of the corresponding signal relative to loading control and basal setting (noted as 1(ref)). Right panel, *Cat* overexpression (log$_{10}$-fold change) in *Tsc2*-deficient MEFs. The asterisks indicate significant difference with two-sided *t*-test ($P = 1 \times 10^{-2}$; replicates/condition $n = 3$, and independent experiments $n = 2$). The bars indicate mean $\pm$ SD. Dotted horizontal lines indicate 2-fold (top) and 0.5-fold (bottom).

G Representative images of immunohistochemical detection of acrolein in LAM lung lesions. Seven LAM patients were analyzed, and assay controls are shown in Appendix Fig S2.

Source data are available online for this figure.

24 hours in 10% FBS media without L-histidine. LC-MS/MS-based quantifications revealed significantly lower levels of labeled histamine in *Tsc2*-null MEFs and, conversely, a trend toward higher levels of the metabolic intermediate N-methylhistamine, and significant overabundance of labeled MIAA (Fig 5B), demonstrating an enhanced metabolic flux in the catabolism of histamine in *Tsc2*-deficient MEFs. In parallel, *Tsc2*-deficient MEFs treated with rapamycin for 16 h in complete media showed a significant increase of histamine relative to the same cells treated with DMSO (Fig 5C). A

recent plasma metabolic study of LAM patients treated with a combination of sirolimus and hydroxychloroquine identified significant reduction of histidine metabolism, among other alterations (Tang *et al*, 2019); however, MIAA did not show changes in this study, which might be due to the drug combination, time course variations and/or the need of targeted quantification.

Cell types with active histamine metabolism are commonly responsive to histamine-mediated signaling and, of the histamine receptors, HRH1 and HRH2 are the most broadly expressed in

human tissue and organs (Schwelberger et al, 2013). HRH1 is also expressed in breast cancer with stem cell-like features and regulates its progression (Fernández-Nogueira et al, 2018). Analyzing the lung-metastatic breast cancer dataset (Minn et al, 2005) that gave rise to the original metabolic predictions (Fig 1A and B) revealed an association between high HRH1 expression and lung metastasis (Appendix Fig S4A). In addition, analyzing data from The Cancer Genome Atlas (Cancer Genome Atlas Network, 2012), the expression profiles of HRH1 and TSC2 were found to be negatively correlated (PCC $= -0.15$, $P = 2 \times 10^{-6}$) and, conversely, HRH1 expression was found to be positively correlated with the canonical lung-metastasis signature (Minn et al, 2005) and mTOR pathway (Appendix Fig S4B).

Following on from the breast cancer predictions, HRH1 was found overexpressed in Tsc2-deficient relative to wild-type MEFs when exposed to serum-reduced (0.5% FBS) media (Fig 5D). In addition, the lung lesions of all seven LAM patients analyzed were clearly positive for HRH1 expression (Fig 5E and Appendix Fig S2). Next, we found that Tsc2-deficient MEFs experienced a larger synergistic inhibitory effect (combination index (CIx) < 1 and $P < 0.05$) when loratadine—an antagonist of HRH1—was combined with mTOR inhibition (Fig 5F). In addition, the viability of Tsc2-deficient MEFs was increased as levels of histamine rose in serum-reduced media without L-histidine (Fig 5G). Consistent with this, Tsc2-deficient MEFs also overexpressed the plasma membrane histamine transporters Slc22a2 and Slc22a3 (Fig 5D and Appendix Fig S5). Furthermore, these cells showed reduced viability in media without L-histidine (Fig 5H). Lastly, coherent with histamine-mediated signaling, the phospho-Tyr783 PLCγ1 signal increased in Tsc2-deficient MEFs exposed to histamine in serum-reduced media without L-histidine (Fig 5I). Collectively, these findings are evidence of active histamine signaling and metabolism occurring in a LAM cell model, and the previous plasma and tissue analyses suggest that these processes occur in patients.

## Targeting histamine metabolism and HRH1 signaling in an immunocompetent LAM tumor model

The function of immune modulators is an important factor in LAM biology (Liu et al, 2018; Maisel et al, 2018). Histamine is pro-inflammatory and its signaling through defined receptors can influence innate and adaptive immune responses (Oldford & Marshall, 2015; Branco et al, 2018). Therefore, we evaluated the potential benefit of targeting HRH1, MAO-A, and MAO-B in tumors produced by syngeneic mouse Tsc2-deficient 105K cells subcutaneously engrafted in C57BL/6J mice (Liu et al, 2018). Compared with TSC2-reconstituted cell counterparts, this LAM model also overexpressed both MAO-A and MAO-B and showed higher basal MAO activity (Appendix Fig S6). Importantly, LC-MS/MS assays of plasma from immunocompetent female mice harboring Tsc2-deficient 105K tumors showed higher levels of MIAA relative to age-matched animals without tumors (Fig 6A). Furthermore, the front of these tumors also frequently exhibited mast cells (Appendix Fig S7).

The selected compounds for the in vivo assays comprised the approved drugs introduced above: clorgyline, MAO-A inhibitor, administered 10 mg/kg/day intraperitoneal; rasagiline, MAO-B inhibitor, administered 1 mg/kg/day intraperitoneal; and loratadine, HRH1 antagonist, administered 25 mg/kg/day oral. In monotherapy, these drugs provided significant inhibition of Tsc2-deficient 105 K tumor growth (Fig 6B). Then, combination of clorgyline or loratadine with rapamycin (0.25 mg/kg/day intraperitoneal) revealed further reduction of tumor growth relative to rapamycin alone (Fig. 6C). Evaluation of tumor weight reduction in these assays showed similar trends, in particular for clorgyline and loratadine in monotherapy, and for loratadine combined with rapamycin (Fig 6D). Since loratadine might interact with rapamycin pharmacokinetics, we quantified rapamycin in mice harboring tumors to which the previous regimens of rapamycin in monotherapy and combined with loratadine had been administered. No significant

---

**Figure 5. Histamine-mediated signaling and metabolism in LAM.**

A   Overabundance of histamine in Tsc2 wild-type relative to Tsc2-deficient MEFs, both growth in DMEM 10% FBS. The asterisks indicate significant difference with two-sided t-test ($P = 0.017$); replicates/condition $n = 5$, and independent experiments ($n = 2$). Average values are indicated with lilac-colored lines.

B   Isotope profiling of [13C$_6$,15N$_3$]-labeled L-histidine measured by LC-MS/MS in Tsc2-deficient and wild-type MEFs grown in DMEM 10% FBS without unlabeled L-histidine. Labeled-measured metabolites are depicted in the X-axis. The significant differences correspond to two-sided t-test (*$P = 0.032$ and **$P = 0.008$; replicates/condition $n = 5$). Average values are indicated with lilac-colored lines.

C   Fold-change variation of histamine and MIAA levels measured by LC-MS/MS assays in rapamycin-exposed (20 nM, 16 h) Tsc2-deficient MEFs relative to DMSO-exposed for the same period. The significant difference corresponds to one-sided t-test ($P = 0.012$; replicates/condition $n = 4$ and independent experiments $n = 2$). Average values are indicated with lilac-colored lines.

D   Western blot results of HRH1, SLC22A3, and loading control from Tsc2-deficient and wild-type MEFs exposed to DMSO or treated with 20 nM everolimus for 16 h in DMEM 10% or 0.5% FBS (independent experiments $n = 2$). The expression levels are indicated by the ratio of the corresponding signal relative to loading control and basal setting (noted as 1(ref)).

E   Representative images of immunohistochemical detection of HRH1 in LAM lung lesions. A total of seven LAM patients were analyzed, and assay controls are shown in Appendix Fig S2.

F   Evaluation of cell viability inhibition by loratadine alone (concentrations shown on X-axis, from 0 to 100 µM) or combined with everolimus (fixed to 20 nM) in Tsc2-deficient and wild-type MEF cultures grown in DMEM 10% FBS for 72 h. The synergistic combination index (CIx < 1) is shown (replicates/condition $n = 3$, independent experiments $n = 3$). Each data point represents the mean and SD.

G   Percentages of viability of Tsc2-dificient MEFs exposed to increasing concentrations of histamine (X-axis) in DMEM 0.5% FBS without L-histidine for 72 h. The asterisk indicates significant difference with one-sided t-test ($P = 0.040$; replicates/condition $n = 4$–6, independent experiments $n = 2$). The bars indicate mean ± SD.

H   Percentages of viability of Tsc2-dificient and wild-type MEFs exposed to DMEM 10% or 0.5% FBS with or without L-histidine for 72 h. The asterisks indicate significant differences with one-sided t-test (10% FBS, $P = 0.017$; 0.5% FBS, $P = 5 \times 10^{-4}$; replicates/condition $n = 4$–6, independent experiments $n = 2$). The bars indicate mean ± SD.

I   Western blot results of phospho-Tyr783 PLCγ1 and loading control in Tsc2-deficient MEFs grown in DMEM 0.5% FBS without L-histidine and exposed to histamine (1 µM) for 10 min or 1 h. The signals of phospho-Tyr783 PLCγ1 relative to loading control or non-phospho PLCγ1 and to basal setting (noted as 1(ref)) are indicated.

Source data are available online for this figure.

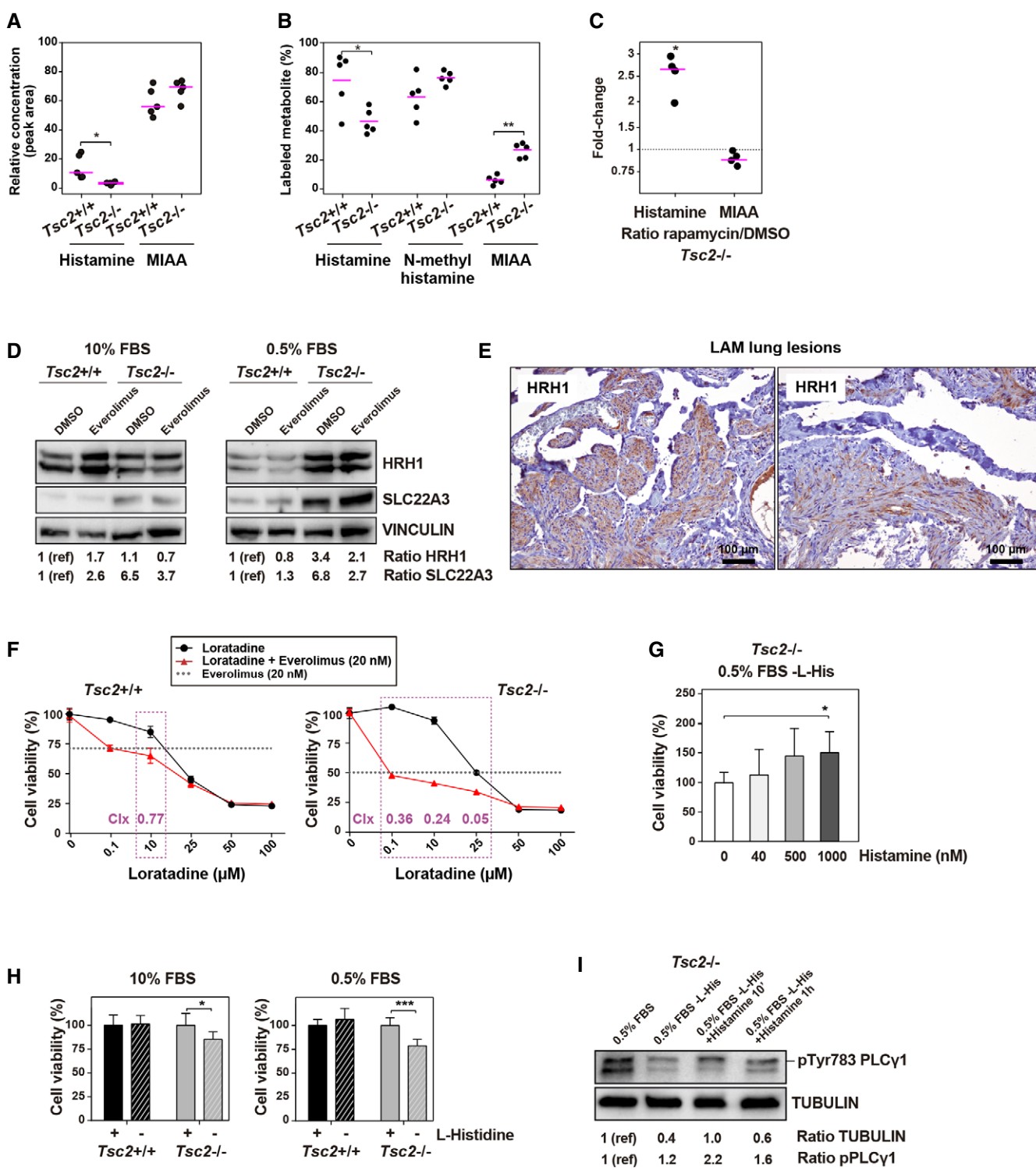

**Figure 5.**

differences in rapamycin levels in blood and tumor samples were detected when loratadine was co-administered (Appendix Fig S8).

Complementary *in vivo* assays were performed to corroborate the implications for histamine metabolism and signaling. Depletion of *Maoa* and *Hrh1* expression with short-hairpin RNA sequences (shRNAs) also revealed tumor growth inhibition, relative to shRNA

control pLKO (Fig 6E and F). The effects were stronger for *Maoa* depletion, but the efficiency of the shRNAs differed (Appendix Fig S9). *In vitro* proliferation assays confirmed the inhibitory effect mediated by depletion of *Maoa* and *Hrh1* expression in *Tsc2*-deficient 105K cells; a synergistic interaction was further suggested in cells exposed to both rapamycin and *Maoa* expression depletion

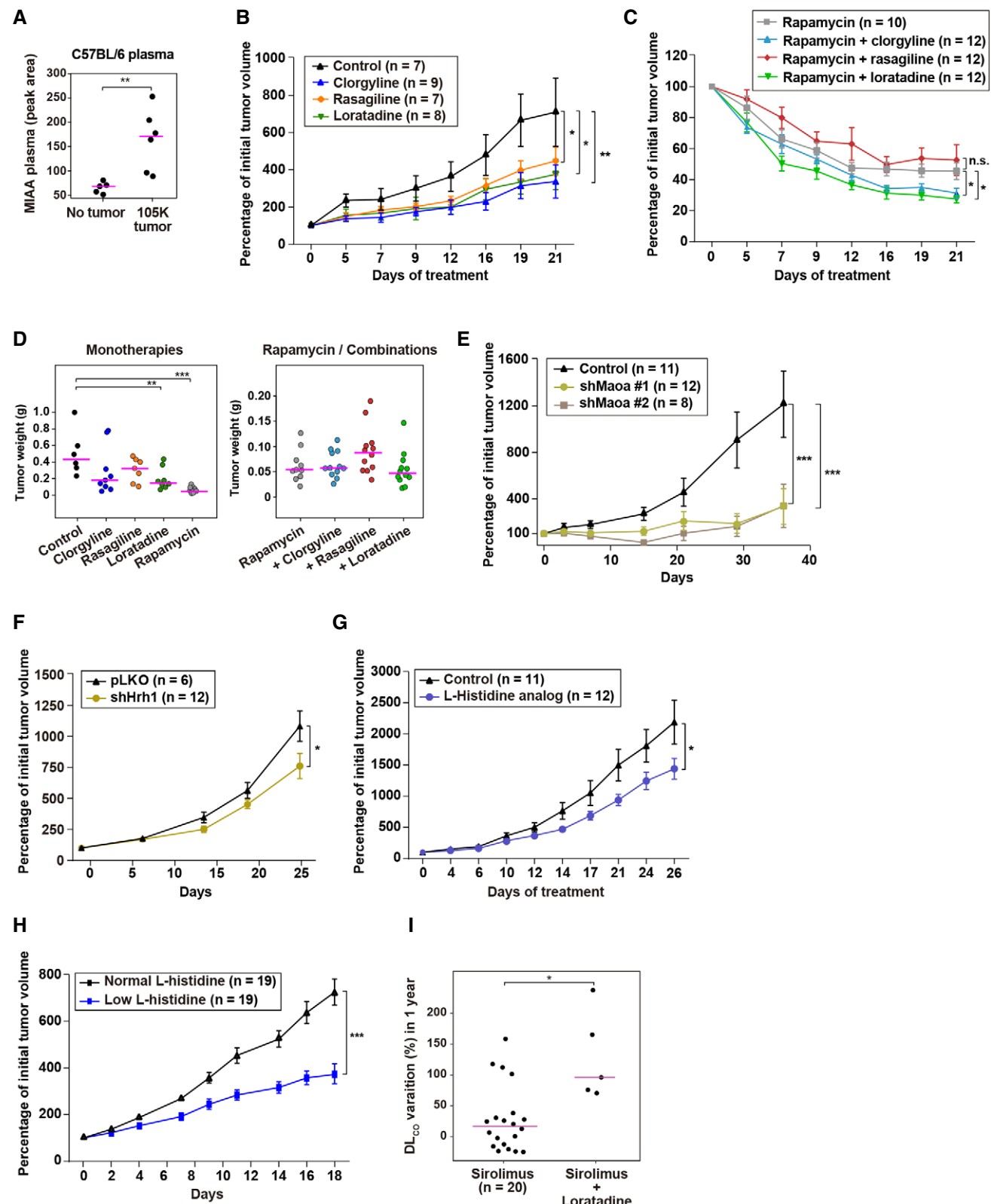

**Figure 6.**

**Figure 6.  Inhibition of LAM tumorigenesis by targeting histamine metabolism and signaling.**

A   LC-MS/MS quantification of MIAA in plasma from C57BL/6 female mice with or without *Tsc2*-deficient 105 K tumors grown. The asterisks indicate a significant difference with two-sided *t*-test ($P = 8 \times 10^{-3}$; replicates/condition $n = 5$). Average values are indicated with lilac-colored lines.

B   Inhibition of *Tsc2*-deficient 105 K tumor growth with different monotherapies, as indicated in the inset. Asterisks indicate significant reductions relative to vehicle (two-way ANOVA; clorgyline $P = 0.015$, loratadine $P = 0.018$, and rasagiline $P = 0.049$). Each data point represents the mean and SEM.

C   Further reduction of *Tsc2*-deficient 105 K tumor growth by rapamycin combined with clorgyline or loratadine, relative to rapamycin alone. Asterisks indicate significant reductions relative to rapamycin alone (two-way ANOVA; clorgyline $P = 0.035$, loratadine $P = 0.045$). Each data point represents the mean and SEM.

D   *Tsc2*-deficient 105 K tumor weight (g) changes at the end of the rapamycin and rapamycin-combination assays. Asterisks indicate a significant reduction relative to control, as determined by a one-sided *t*-test (loratadine $P = 0.009$ and rapamycin $P = 3 \times 10^{-5}$). Average weight values are indicated with lilac-colored lines.

E   Inhibition of *Tsc2*-deficient 105 K tumor growth with *Maoa* expression depletion using shRNAs. Asterisks indicate significant reductions relative to control pLKO (two-way ANOVA; $P = 1 \times 10^{-3}$). Each data point represents the mean and SEM.

F   Inhibition of *Tsc2*-deficient 105K tumor growth with *Hrh1* expression depletion. Asterisk indicates significant reduction relative to control pLKO (two-way ANOVA; $P = 0.035$). Each data point represents the mean and SEM.

G   Inhibition of *Tsc2*-deficient 105 K tumor growth with administration of α-methyl-DL-histidine. Asterisk indicates significant reduction relative to vehicle (two-way ANOVA; $P = 0.028$). Each data point represents the mean and SEM.

H   Inhibition of *Tsc2*-deficient 105K tumor growth by the administration of a low L-histidine concentration mouse diet (0.07% versus 0.49%). Asterisks indicate significant reduction relative to 0.49% L-histidine diet (two-way ANOVA; $P = 0.001$). Each data point represents the mean and SEM.

I    Increase in $DL_{CO}$ over a year in patients treated with rapamycin plus loratadine, relative to those treated with rapamycin alone. Significant difference determined by a two-sided Mann–Whitney test ($P = 0.045$; number ($n$) of samples are indicated). Average values are indicated with lilac-colored lines.

Source data are available online for this figure.

(Appendix Fig S10A). Analogous *in vivo* assays confirmed significant tumor growth reduction when the cells were depleted in *Maoa* or *Hrh1* expression and in mice treated with rapamycin, relative to pLKO-transduction and equivalent administration of rapamycin (Appendix Fig S10B). In addition, systemic administration of an L-histidine analog (α-methyl-DL-histidine, 150 mg/kg/day intraperitoneal) also reduced tumorigenesis (Fig 6G). Moreover, there was a similar consequence to administering a mouse diet with a relatively low concentration of L-histidine: 0.07% versus a normal concentration of 0.49% (Fig 6H). The low L-histidine diet caused just a 5% weight loss relative to the standard diet.

Loratadine is a widely used second-generation H1 antihistamine (Church & Church, 2011). Data about the use of this drug were compiled in parallel with this study in a LAM cohort from Poland (Appendix Table S4). Loratadine users were defined as those patients who were medicated with 10 mg/day of the drug for at least 10 months in a year at the same time as receiving rapamycin treatment. Using linear mixed-effects regression models, we compared the percentage of variation in pulmonary function test measures over a year in patients with rapamycin alone versus those that also used loratadine. Data for $FEV_1$ did not indicate differences. However, $DL_{CO}$ suggested greater improvements in patients who were also medicated with loratadine (Fig. 6I).

## Histopathological changes associated with LAM tumor inhibition

Evaluation of candidate molecular alterations underlying tumor inhibition revealed significantly lower HRH1 levels in the loratadine-treated tumors, and higher levels of the mitochondrial fission marker phospho-Ser616 DRP1 in the rasagiline-treated tumors, with a similar trend in the two other monotherapy settings (Fig 7A). Enhancement of mitochondrial fission has been linked to

**Figure 7.   Molecular and histopathological changes associated with targeting histamine metabolism and signaling.**

A   Top panels, Western blot results of HRH1 and phospho-Ser616 DRP1 expression in vehicle control and single-drug treated tumors. Bottom panels show quantifications; significant differences correspond to one-sided *t*-test (*$P = 0.011$ and **$P = 2 \times 10^{-3}$; tumors/group $n = 4$). Average values are indicated with lilac-colored lines.

B   Western blot results showing increased HRH1 expression in combinations relative to rapamycin alone, reduced expression of S6 total in the rapamycin–loratadine combination, and raised phospho-Ser616 DRP1 expression in this combination. Bottom panels show quantifications and significant differences with one-sided *t*-test (HRH1, clorgyline $P = 0.011$, loratadine $P = 0.049$, rasagiline $P = 0.021$; total S6, loratadine $P = 4 \times 10^{-4}$; phospho-Ser616 DRP1, loratadine $P = 0.037$; tumors/group $n = 4$). Average values are indicated with lilac-colored lines.

C   Graph showing the percentages of MitoSOX red-positive *Tsc2*-deficient 105 K cells treated for 24 h with DMSO or drugs *in vitro* with 10% FBS complete medium (except for the condition without L-histidine). Clorgyline 1 μM, loratadine 100 nM, rasagiline 1 μM, and rapamycin 20 nM. Differences relative to rapamycin alone were determined by a two-sided *t*-test (DMSO $P = 6 \times 10^{-5}$, clorgyline $P = 2 \times 10^{-5}$, loratadine $P = 6 \times 10^{-4}$, rasagiline $P = 4 \times 10^{-5}$; combination with clorgyline $P = 1 \times 10^{-4}$, loratadine $P = 3 \times 10^{-5}$, and rasagiline $P = 9 \times 10^{-4}$; replicates/condition $n = 5$; assays $n = 2$). The bars indicate mean ± SD.

D   Results of blind histopathological evaluation of *Tsc2*-deficient 105 K tumors treated with vehicle or drugs, in monotherapy or in combination with rapamycin. The histograms show the proportion of defined phenotypic scores: 1+, < 5% of the tumor; 2+, 5–50% of the tumor; and 3+, > 50% of the tumor. Cytological atypia was graded as mild (1+), moderate (2+), or severe (3+). Significant differences relative to rapamycin were determined using Fisher's exact test (epithelioid ***$P = 9 \times 10^{-4}$ and *$P = 0.025$; fibrosis, clorgyline and loratadine *$P = 0.015$, and rasagiline *$P = 0.032$; glandular, clorgyline *$P = 0.030$ and rasagiline *$P = 0.045$; and atypia, loratadine $P = 0.010$).

E   Representative images of hematoxylin–eosin-stained *Tsc2*-deficient 105K tumors treated with vehicle or monotherapies. Tumors treated with rapamycin tend to have a fascicular growth pattern with bundles of spindle cells and foci of fibrosis, whereas most of the tumors of the other treatment groups showed extensive epithelioid morphology. Scale bars are shown.

F   Representative images of hematoxylin–eosin-stained *Tsc2*-deficient 105K tumors treated with rapamycin alone or rapamycin combinations. With the addition of a second drug to rapamycin, tumors more frequently tended to show glandular differentiation and less atypia. Scale bars are shown.

Source data are available online for this figure.

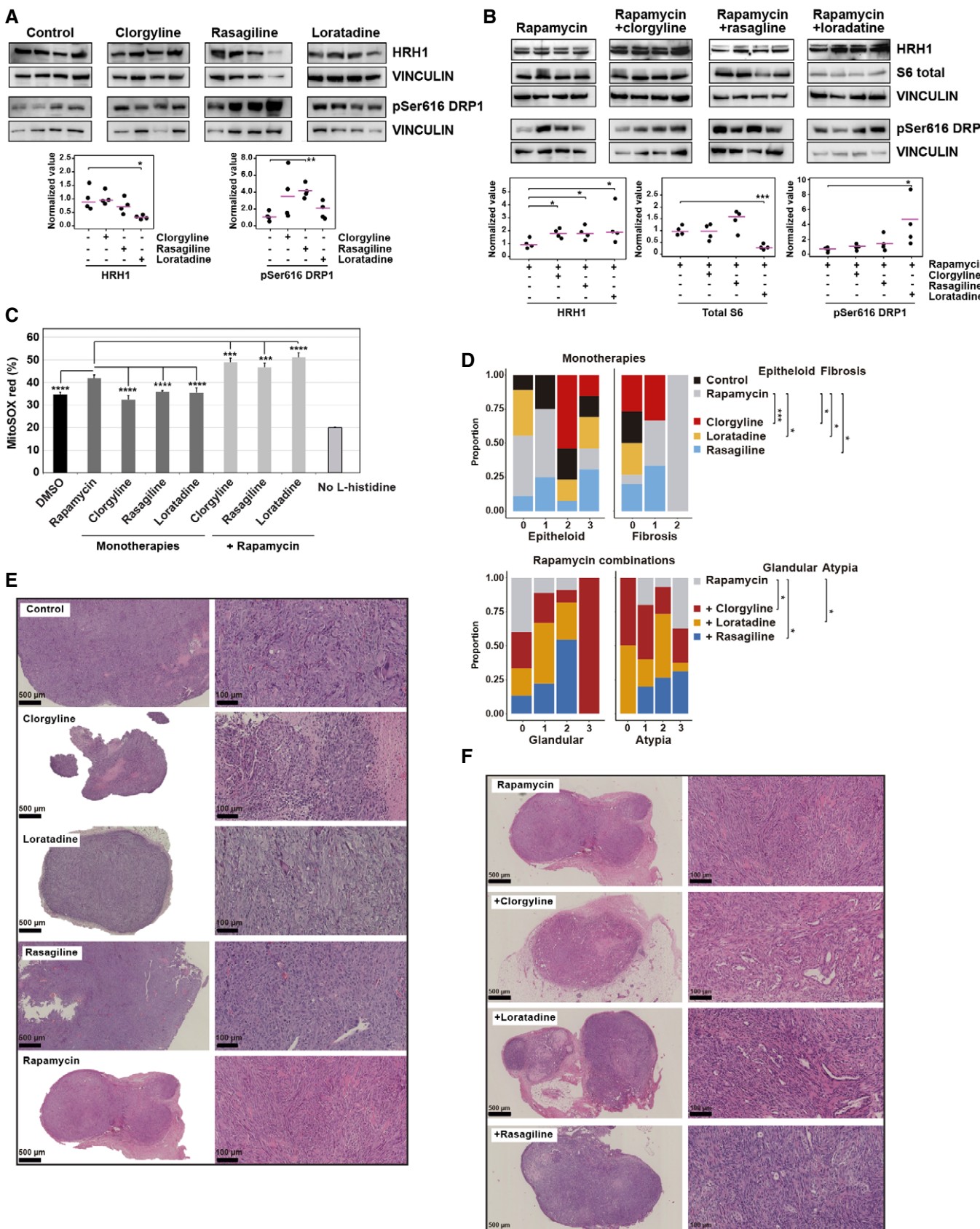

**Figure 7.**

suppressed breast cancer metastasis (Humphries et al, 2020). Rapamycin combinations also altered histamine-mediated signaling, as indicated by increased HRH1 levels relative to rapamycin alone (Fig 7B), and synergistically increased mitochondrial ROS, as measured by MitoSOX assays in vitro (Fig 7C). In addition, the combination of rapamycin and loratadine reduced the total levels of ribosomal protein S6, while also increased phospho-Ser616 DRP1 (Fig 7B). In parallel, we noted that the in vitro sensitivities of NCI-60 cancer cell lines to rapamycin and loratadine were positively correlated independently of cancer type (Appendix Fig S11). We analyzed the levels of selected phospho-sites and total expression of AKT and ribosomal S6 protein in Tsc2-deficient MEFs and 105 K cells exposed in vitro to vehicle, monotherapies, and rapamycin combinations, for 5 h in complete media. The monotherapies of clorgyline, loratadine, and rasagiline showed relative reductions in both phospho-Ser235/236 and total S6, although to a lesser extent than rapamycin (Appendix Fig S12).

The three monotherapies and rapamycin combinations did not cause enhanced tumor cell death (Appendix Fig S13 and Table S5) or alteration of autophagy in vitro (Appendix Fig S14), but tumor cell proliferation decreased as measured by Ki67 immunostaining (Appendix Fig S15). Histopathological evaluation revealed features of phenotypic differentiation linked to LAM inhibition: tumors treated with vehicle or rapamycin tended to have a fascicular growth pattern with bundles of spindle cells and foci of fibrosis, whereas the three other single-drug treatments showed extensive epithelioid morphology (Fig 7D and E, and Appendix Table S5). RNA sequencing of tumors treated with vehicle, rapamycin and/or loratadine further indicated frequent changes in processes linked to cell differentiation and development with exposure to loratadine (Appendix Fig S16). In addition, the rapamycin combinations caused tumors with higher glandular differentiation (Fig 7F and Appendix Table S5). In accordance with the effects of drugs, histopathological evaluation of tumors from the assays using Maoa/Hrh1 shRNAs, L-histidine analog, and low L-histidine diet also commonly showed epithelioid morphology (Appendix Figs S17 and S18). To specifically assess diseased cell changes, we transduced Tsc2-deficient 105K cells in vitro with the specific shRNAs and quantified the expression of informative markers relative to pLKO control. The level of the epithelial marker Epcam increased, while those of the mesenchymal markers (Fn1, Snai1, Vim, and Twist1) decreased as the expression of Maoa or Hrh1 was depleted (Appendix Fig S19). Next, we evaluated the invasion and migration capacity of Tsc2-deficient 105K cells under the above conditions and observed a significant decrease in both competences with the depletion of Maoa or Hrh1 expression, relative to pLKO control (Appendix Fig S20).

## Discussion

This study proposes novel therapeutic opportunities and their associated blood biomarkers in LAM. The therapies are centered on targeting histamine metabolism and signaling and have been assessed using an immunocompetent LAM cell model employing complementary strategies: approved drugs, gene expression depletion (Maoa and Hrh1), metabolite analog (α-methyl-DL-histidine), and a specific diet (low L-histidine concentration). The proposed

therapies provide disease control as single agents or, in some instances, may be more beneficial when combined with rapamycin. MIAA, the major product of histamine metabolism, can be combined with VEGF-D to reinforce a differential diagnosis, and measuring histamine levels can complement the estimates of disease burden and pulmonary function. Analysis of retrospective clinical data suggests some functional benefit from the combination of rapamycin and loratadine; however, a multicenter phase II clinical trial will be initiated in Spain to formally assess the safety and initial benefit of this combination in LAM patients (EudraCT reference 2020-000702-29). In parallel to this study, recent evidence has shown an interplay between LAM, fibroblast, and mast cells in lung nodules promoting disease progression, which may be inhibited by blocking mast cell activation (Babaei-Jadidi et al, 2021). Mast cells were previously shown to be markedly present in LAM lung nodules (Valencia et al, 2006) and were also evident in the analysis of single-cell LAM profiles (Guo et al, 2020). Collectively, mast cell biology may emerge as a central target for improving LAM management and monitoring.

Our data indicate that targeting histamine signaling and metabolism can contribute to disease control by affecting the pathological phenotype. Cancer stem cells are characterized by the relative overexpression and activity of ALDH isoforms, a metabolic trait associated with the metastatic potential of different neoplasms, including breast cancer (Ginestier et al, 2007). MAOA is overexpressed in the transition from breast cancer bone metastatic dormancy to relapse (Lu et al, 2011) and is required for breast tumor-initiating cells (Gwynne et al, 2019) and promotes prostate cancer metastasis (Wu et al, 2017). MAOB was recognized in a gene signature that characterizes mammary stem cells (Lim et al, 2010), and its expression is a prognostic factor in breast cancer (Cha et al, 2018). In addition, as noted earlier, MAOB and ALDH genes are included in a recently described single-cell LAM[core] signature (Guo et al, 2020). Therefore, our current observations in LAM-related cell models, tissue, and plasma are consistent with a pivotal role for ALDHs and MAOs in LAM diseased cells and may extend the concept of LAM as a stem cell-like disease (Henske & McCormack, 2012; Ruiz de Garibay et al, 2015; Pacheco-Rodríguez et al, 2019). In this scenario, the role of MAO-B may be relatively more relevant as it is specifically involved in histamine degradation (Schwelberger et al, 2013). However, the identification of additional monoamine-derived metabolites in LAM plasma, and the results from the assays with clorgyline and MAO-A depletion, suggest that this isoform also has a substantial role in disease biology. As both isoforms are expressed in LAM models and lung lesions, they may be co-regulated, as occurs in other tissues and conditions (Shih et al, 2011). The predictable crosstalk between monoamine metabolism, histamine, and mTOR signaling could be linked to sensing amino acid levels (Schwelberger et al, 2013; Saxton & Sabatini, 2017), and converge on ribosomal S6 activation, as described in other cellular settings (Dickenson, 2002). However, further studies are needed to determine the underlying mechanism of histamine-mTOR signaling crosstalk.

The biomarker predictions that underpinned this study highlight the merit of undertaking further analyses of identified metabolites, and of strategies targeting histamine signaling and metabolism, in different neoplasms, and especially in metastatic breast cancer. Active histamine synthesis and metabolism are established factors in normal and cancer breast tissue (Reynolds et al,

1998; von Mach-Szczypiński *et al*, 2009) and may be influenced by hormonal status (Kierska *et al*, 1997). Derived metabolites have been recognized in plasma from breast cancer patients (von Mach-Szczypiński *et al*, 2009). HRH1 is also known to function in normal and cancer breast cell lines and tissue (Davio *et al*, 1995). In parallel, accumulation of mast cells—major source of histamine—in the tumor microenvironment is correlated with poor prognosis, increased metastasis, and reduced survival of several cancer types, including that of the breast (Maciel *et al*, 2015). Abnormally enhanced mast cell activity has been associated with an increased risk of breast cancer and of melanoma and cervical cancer (Molderings *et al*, 2017). Recent epidemiological reports from the Swedish Cancer Registry suggest that medication with loratadine, or its major metabolite, desloratadine, substantially improves breast cancer survival (Fritz *et al*, 2020). While breast cancer is a complex disease including subtypes that differ in their metastatic preference and hormonal influence, among other factors, these observations appear to be consistent with the results obtained for LAM in this study.

Together, the results of our study have identified potential LAM biomarkers for differential disease diagnosis and monitoring, which in turn are connected to promising therapies using approved drugs. Among the tested drugs, loratadine shows beneficial effects in monotherapy and combined with rapamycin and is well tolerated and usually safe, with many individuals in the general population receiving this medication for common conditions. The proposed biomarkers and therapies are supported by molecular, metabolic, cellular, and tissue analyses that depict active histamine metabolism and signaling in LAM.

# Materials and Methods

## Patients and samples

### Spain

Lymphangioleiomyomatosis patients were recruited and lung tissue samples collected by participating centers (International LAM Clinic, University Hospital Vall d'Hebron; University Hospital La Princesa; University Hospital Clínica Puerta del Hierro; Hospital Clínic de Barcelona; University Hospital Virgen del Rocío; and University Hospital of Bellvitge) and with the support of the Spanish LAM Association (AELAM). Blood samples were collected during the 2017 and 2018 annual AELAM patient conferences, so the time between undertaking pulmonary function tests and sample acquisition varied, making it impossible to assess the former relative to plasma metabolites. The data collected consisted of age at diagnosis, age at sample extraction, diagnosis of AML, chylothorax, pneumothorax, TSC, and therapy used at the time of sample extraction. All patients provided written informed consent, and the study was approved by the ethics committees of IDIBELL and the Instituto de Investigación Sanitaria La Princesa, Hospital de Henares, Spain. Control samples were obtained from healthy pre-menopausal women from a similar age distribution to that of the LAM patients.

### UK

The LAM cohort was recruited at the National Centre for LAM in Nottingham between 2011 and 2018. The East Midlands Research Ethics Committee approved the study (13/EM/0264), and all participants gave written informed consent. Medical history, CT scans of the chest and abdomen, and full lung function tests were performed when patients were recruited to the study as part of their clinical care. $FEV_1$ and $D_{LCO}$ were also measured at each follow-up visit. The disease burden score was used to stratify the whole-body burden of LAM. For each subject, one point is given for each of (i) more severe lung disease defined by an $FEV_1$ or $DL_{CO}$ of < 60% predicted, (ii) the presence of an AML at the time of assessment, and (iii) the presence of lymphatic involvement visible on imaging (lymphadenopathy, cystic lymphatic mass, or chylous effusion), resulting in a score of 0–3. Control samples corresponded to healthy women with no history of lung disease with a similar age distribution to that of the LAM patients.

### Poland

The LAM cohort was recruited at the National Tuberculosis and Lung Diseases Research Institute. The local Bioethics Committee approved the study and patients signed informed consent. In addition to standard clinical follow-up, data collected for the years 2010–2019 included loratadine use (dose and period), smoking status (packs/year), and diagnosis of allergy, AML, asthma, chylothorax, lymphangioma, perivascular epithelioid cell tumor, pneumothorax, and TSC. The average follow-up was 4.5 years, and pulmonary function tests were performed at initiation of rapamycin (sirolimus), after 3, 6, and 12 months, and subsequently every year. The tests were performed according to the joint guidelines of the American Thoracic Society and European Respiratory Society, and lung volumes measured by body plethysmography (Jaeger MasterScreen software version 4.65) and $DL_{CO}$ using the single breath technique.

### LAM-related pulmonary diseases

Emphysema-related plasma samples were collected from patients attending the Interstitial Lung Disease Unit of the University Hospital of Bellvitge. These were adult individuals (31–59 years old) with a smoking history that led to them present dyspnea on exertion, but whose $FEV_1/FVC$ ratio was less than 0.70. According to established guidelines, diagnosis of chronic obstructive pulmonary disease (COPD) requires the presence of three features (symptoms, $FEV_1/FVC < 0.70$, and smoking or other noxious exposure), so COPD was excluded from the search for emphysema cases in this study. One of the cases also presented rheumatoid arthritis (higher predisposition to emphysema even with little tobacco exposure). The plasma samples of patients with Langerhans cell histiocytosis, Sjögren syndrome, and systemic lupus erythematosus were collected by the ILD Center of Excellence, St. Antonius Hospital Biobank, Nieuwegein, The Netherlands. The study was approved by the St. Antonius Hospital ethics committee (reference R05-08A), and all participants provided written informed consent. Patients with lupus erythematosus were included in this study because it is the most common connective tissue disease affecting the lung, can cause inflammation and fibrotic manifestations similar to those of interstitial lung disease and might exhibit biomarker overlap (Lamattina *et al*, 2018).

The studies conformed to the principles set out in the WMA Declaration of Helsinki and the Department of Health and Human Services Belmont Report.

## Public gene expression data

Metastatic breast cancer data were taken from the corresponding publication (Minn et al, 2005) and Gene Expression Omnibus (GEO) reference GSE5327. This original dataset from the Memorial Sloan-Kettering Cancer Center included 82 patients with 12 lung metastasis events (Minn et al, 2005). Analysis of these data revealed an association between low TSC2 expression and lung metastasis (Ruiz de Garibay et al, 2015). The dataset was then used to identify enzyme-coding genes whose expression profiles are negatively correlated with TSC2. Official gene names and Entrez IDs were cross-referenced to UniProt enzymatic protein identifiers (human only and manually curated). Associated metabolites from a genome-scale metabolic network model (Recon 2.2 (Swainston et al, 2016)) were mapped to enzymes, resulting in a total of 47 metabolites, which were evaluated for overrepresented pathways using the MetaboAnalyst (Chong et al, 2018) tool with standard parameters. Normalized breast cancer RNA-seq data from The Cancer Genome Atlas were obtained from cBioPortal, and the GSEA method was applied with standard parameters (Subramanian et al, 2005).

## Gene expression quantification

For semi-quantitative gene expression analyses, total RNA was isolated from cells or tissues using TRIzol reagent (Thermo Fisher) and complementary DNA (cDNA) synthesized using the High Capacity cDNA Reverse Transcription Kit (Thermo Fisher), following the manufacturer's protocol. Gene expression values were measured using reverse-transcription polymerase chain reactions with SYBR Green (Applied Biosystems). The mouse control genes were Actb and Ppia, and differences were computed by the $\Delta\Delta Ct$ method. The primer sequences used in these assays are listed in Appendix Table S6.

## RNA-seq analysis

Total RNA from tumors was extracted using TRIzol (Thermo Fisher Scientific) and, following quality controls, single-end-sequenced at the IRB's facility in Barcelona. The RNA-seq reads were trimmed for adaptors, masked for low-complexity and low-quality sequences and subsequently quantified for transcript expression using Kallisto v0.43.16 (Bray et al, 2016) and mouse genome version mm9. Gene-level quantification was carried out using the tximport (Soneson et al, 2015) Bioconductor package, mm9, and Ensembl v94 annotations. Differential expression was analyzed using DESeq2 v2.13 (Love et al, 2014). GO term enrichment was analyzed using clusterProfiler (Yu et al, 2012) and GOnet (Pomaznoy et al, 2018).

## LC-MS/MS assays and model analysis

Plasma samples (50 µl) were mixed with 300 µl cold methanol:water, vortexed, and cold-centrifuged. Cell pellets were scrapped, collected, and frozen. The pellets were resuspended in cold methanol:water and metabolite lysates purified with three rounds of liquid $N_2$ immersion and sonication, followed by 1 h on ice before centrifugation. For the cell culture assays, 100 µl of media was lyophilized and resuspended in cold methanol:water. The extracts were analyzed by an ultra-high performance LC system coupled to a 6490 triple-quadrupole mass spectrometer (QqQ, Agilent Technologies) with electrospray ion source (LC-ESI-QqQ) working in positive mode. Plasma extract (2 µl) and cell and medium extract (1 µl) were injected into the LC system. An ACQUITY UPLC HSS T3 column (1.8 µm, 2.1 × 150 mm, Waters) and a gradient mobile phase consisting of water with 0.1% formic acid (phase A) and acetonitrile with 0.1% formic acid (phase B) were used for chromatographic separation. The gradient was as follows: isocratic for 2 min at 10% B, 2–3 min rising to 80% B, 30 s maintaining 80% B, up to 4 min until the percentage of B decreased to 10%, and finally equilibrating the column at 10% B for 7 min. The flow rate of the method was 0.3 ml/min. The mass spectrometer parameters were as follows: drying and sheath gas temperatures of 120°C and 400°C, respectively; source and sheath gas flows of 19 and 12 l/min, respectively; nebulizer flow 20 psi; capillary voltage 2,100 V; nozzle voltage 500 V; and iFunnel HRF and LRF 150 and 60 V, respectively. The QqQ worked in MRM mode using defined transitions. The [13C6,15N3]-labeled L-histidine was purchased from Cambridge Isotope Laboratories (CNLM-758-PK). For these assays, cells were cultured in DMEM 10% FBS for 16 h, washed twice with PBS, and then cultured in 10% FBS medium without L-histidine but with labeled L-histidine (concentration of 42 mg/l). After 24 h, cells were collected in 50 µl of methanol:water (1:1) using a scraper and frozen in liquid nitrogen until use in LC-MS/MS assays. The transitions for unlabeled metabolites and the collision energy (CE(eV)) were as follows: histamine, 112 → 95(12), 112 → 41(32), 112 → 56(36); N-methylhistamine, 126 → 109(12), 126 → 68(24), 126 → 41(28); and MIAA, 141 → 95(12), 141 → 42(44), 141 → 68(36). For labeled metabolites were as follows: histamine, 120 → 102(12), 120 → 44(32), 120 → 60(36); N-methylhistamine, 134 → 116(12), 134 → 73(24), 134 → 44(28); and MIAA, 148 → 101(12), 148 → 44(44), 148 → 73(36). To categorize the best predictive model with fewer metabolite variables, a stepwise backward–forward procedure with Akaike information criterion was applied using the stepAIC function in R software (Venables & Ripley, 2002). Sirolimus concentration in mouse whole blood and tumor samples was determined using a previously validated method based on ultra-high-performance liquid chromatography coupled to tandem mass spectrometry (UHPLC-MS/MS) (Rigo-Bonnin et al, 2015). The method was initially developed for human whole blood samples and subsequently adapted and validated for mouse whole blood and other biological fluids.

## Immunohistochemistry, immunohistofluorescence, and TUNEL assays

The assays were performed on serial paraffin sections using an EnVision kit (Dako). Antigens were retrieved using citrate-based (pH 6) or EDTA-based (pH 9) buffers. Endogenous peroxidase was blocked by pre-incubation in a solution of 3% $H_2O_2$ performed in 1x phosphate-buffered saline with 10% goat serum. Slides were incubated overnight at 4°C with primary antibody diluted in blocking solution. Secondary anti-mouse or anti-rabbit peroxidase-conjugated antibodies (Envision+ system-HRP, Dako) were used. Sections were hematoxylin-counterstained and examined with a Nikon Eclipse 80i microscope. For immunohistofluorescence, the slides were incubated with a mix of the two primary antibodies (anti-α-SMA and anti-MAO-A/B). In this case, the secondary antibodies used were

goat anti-mouse Alexa-488 and goat anti-rabbit Alexa-546 (dilution 1:1,000; Thermo Fisher). Sudan black staining was performed to avoid paraffin autofluorescence. The sections were counterstained with DAPI, mounted with coverslips in Fluoromount® Aqueous Mounting Medium (Sigma), and visualized in a Fluorescence DM6000 microscope (Leica Microsystems). Apoptosis in paraffin-embedded tumor tissue sections was evaluated using the DeadEnd™ Colorimetric TUNEL System (Promega).

## Cell lines and media

This study used LAM cell models provided by Prof. Elisabeth P. Henske (Center for LAM Research and Clinical Care, Brigham and Women's Hospital, Harvard Medical School, Boston, USA), which have been described previously: MEFs derived from *Tsc2/Trp53* or *Trp53* knockout mice; and *Tsc2*-deficient and *TSC2*-reconstituted mouse kidney cystadenoma cells (105 K) cells (Zhang *et al*, 2003; Parkhitko *et al*, 2014). The cell cultures were based on DMEM media supplemented with 10% FBS (Gibco). When indicated, serum-reduced (i.e., 0.5% FBS) medium was assayed. DMEM medium without L-histidine was purchased from US Biological (catalog D9801-02). All cells were grown at 37°C in a humidified atmosphere with 5% $CO_2$, and mycoplasma was tested by PCR every month.

## Cell-based assays

The number of independent assays is indicated in the corresponding figure legend. MAO activity was monitored using a radiometric assay with $^{14}$C-tyramine hydrochloride as the substrate. Data were normalized for protein content and rates expressed as disintegrations of 14C/min/μg protein. ALDH activity was determined using an ALDEFLUOR assay kit (STEMCELL Technologies) following the manufacturer's protocol and analyzed with a Gallios flow cytometer (Beckman Coulter). MitoSOX Red (Thermo Fisher) was used to quantify mitochondrial superoxide production. Cells seeded in 6-well plates were incubated with 5 μM of MitoSOX dye diluted in Hank's Buffered Salt Solution (HBSS) for 10 min at 37°C. Staining was then measured by flow cytometry. The MitoTracker Red CMXRos (Thermo Fisher) was used to measure mitochondrial membrane potential. Cells seeded in 6-well plates were incubated with 500 nM MitoTracker dye diluted in PBS for 30 min at 37°C, and the staining was analyzed by flow cytometry. Assays were performed in duplicate for each condition. $H_2O_2$ production was measured with the Fluorescent Hydrogen Peroxide Assay Kit (Sigma-Aldrich), following the manufacturer's protocol. Briefly, different numbers of cells were diluted in 50 μl of assay buffer and seeded in a 96-well plate. A standard curve ranging from 0 to 10 μM $H_2O_2$ was prepared. Standards and samples were incubated with red peroxidase substrate, horseradish peroxidase, and assay buffer to a final volume of 100 μl. The plate was incubated for 30 min at room temperature in darkness before measuring the fluorescence intensity ($\lambda_{ex}$ = 540 / $\lambda_{em}$ = 590 nm). The high-resolution Oxygraph-2K (Oroboros Instruments) was used to measure the oxygen consumption in cells. In brief, a concentration of $10^6$ cells/ml diluted in medium was placed in the closed respirometer chamber to measure basal oxygen consumption. Oxygen flux was recorded constantly using DatLab software 4.3 (Oroboros Instruments), and the values obtained were directly proportional to the oxygen consumption. The

histamine-cell viability assays were performed as follows: six replicates of 1,000 cells for each condition were distributed in 96-well plates in 100 μl DMEM 10% FBS without L-histidine. After 12 h, the medium was changed to DMEM 0.5% FBS without L-histidine and without histamine. Fresh histamine was added at the corresponding final concentrations and the medium replaced (DMEM 0.5% FBS without L-histidine, and histamine as appropriate) daily for three days. Cell viability was measured using a CellTiter-Glo Luminescent Cell Viability Assay (Promega) with an EnSpire Multimode Plate Reader. Generally, for other viability assays, cells were plated into 96-well plates (500–2,500 per well) in 4–6 replicas and treated for 72 h. All assays were repeated at least three times and, to avoid assay-specific limitations, measured at least once with both with 2,3-bis-(2-methoxy-4-nitro-5-sulfophenyl)-2H-tetrazolium-5-carboxanilide (XTT, Sigma-Aldrich) and CellTiter-Glo Luminescent Cell Viability Assay (Promega). The results were evaluated relative to control solution-treated cells. The Chou-Talalay (Chou, 2010) method in ComboSyn was used to identify synergistic drug combinations. The CYTO-ID® Autophagy detection kit (Enzo) was used according to the manufacturer protocol with cells grown in complete medium with 10% FBS (except for a condition without L-histidine) and exposed to defined drugs for 24 h (clorgyline 1 μM, loratadine 100 nM, rasagiline 1 μM, and rapamycin 20 nM). In vitro invasion assay. The transwell migration assays were performed as follows: cells were serum-deprived for 24 h, detached, resuspended in serum-free medium and counted; 24-well transwell inserts with 8-μm diameter pore membranes (Sarstedt) were pre-coated with growth factor-reduced Matrigel (Corning) following the manufacturer's instructions; $5 \times 10^4$ cells/well were seeded in triplicate in the upper chamber using 200 μL of serum-free medium, and 500 μL of complete medium was added to the lower chamber as attractant at the same time; after incubation for 24 h at 37°C, cells remaining on the upper surface of the membrane were removed with cotton swabs, and invaded cells on the lower surface of the membrane were fixed with methanol and stained with hematoxylin solution; the cells that passed through the filter were photographed with a bright-field microscope and counted. The wound-healing assays were performed as follows: Cells were seeded in a culture-insert (IBIDI) at a density of $2.5 \times 10^4$ cells/well; after allowing cells to attach overnight, the inserts were removed and cultures washed with PBS to remove non-adherent cells; the cells were then cultured with fresh serum-free medium and photographed at different time points using an inverted microscope; the rate of wound closure was quantified using TScratch (Gebäck *et al*, 2009). The NCI-60 drug response data were downloaded from the Developmental Therapeutics Program repository of the National Cancer Institute (Shoemaker, 2006).

## Compounds, antibodies, and diet

Clorgyline was purchased from Sigma-Aldrich and everolimus, loratadine, rapamycin, and rasagiline were purchased from Selleck Chemicals. The α-methyl-DL-histidine was purchased from ABCR GmbH. The origin, application, and dilution of the primary antibodies are specified in Appendix Table S7. Custom irradiated mouse diets were purchased from International Product Supplies (IPS), had 8% fat and 15% sucrose, and only differed in L-histidine concentration: 0.07% or 0.49%.

**The paper explained**

**Problem**

Lymphangioleiomyomatosis (LAM) is a rare multisystem disease characterized by progressive cystic lung destruction. Rapamycin (sirolimus) is the current standard of care, but this treatment is not sufficiently tolerated in all patients and some cases experience continuous decline of lung function despite of therapy. High level of vascular endothelial growth factor D (VEGF-D) in blood plasma supports differential LAM diagnosis and clinical monitoring, but VEGF-D shows individual variability of unclear origin.

**Results**

Here, we identify LAM plasma biomarkers that lead to preclinical demonstration of novel therapeutic approaches for the disease. The major histamine metabolite, methylimidazoleacetic acid (MIAA), is relatively more abundant in blood plasma of LAM patients, and MIAA quantification may be combined with VEGF-D measure to improve diagnosis. High levels of histamine in LAM plasma are associated with poorer lung function. Studies in LAM cell models depict the disease hallmarks of active histamine signaling and metabolism. In mouse studies, LAM can be inhibited using approved drugs that target the exposed histamine-centered processes. Loratadine, a common antihistamine, is beneficial in monotherapy and may synergize with rapamycin.

**Impact**

This study provides novel insight into disease biology by depicting active histamine metabolism and signaling, which can be exploited to implement additional biomarkers and therapies. A phase II clinical trial has been initiated to assess the combination of rapamycin and loratadine.

## Western blot

All assays were performed on at least three separate occasions. Cells or tissues were lysed in a radioimmunoprecipitation assay (RIPA) buffer supplemented with protease inhibitor (cOmplete, Sigma-Aldrich) and phosphatase inhibitor (PhosSTOP, Sigma-Aldrich) cocktails. Protein concentrations were measured using the Thermo Scientific Pierce BCA Protein assay. Lysates were resolved in sodium dodecyl sulfate-polyacrylamide electrophoresis gels and transferred to polyvinylidene difluoride (PVDF) membranes (Sigma-Aldrich). Target proteins were detected using the primary antibodies listed above, the secondary antibodies purchased from Santa Cruz Biotechnologies (sc-2357 and sc-2005, 1:2,000), and by chemiluminescence detection with the Nove ECL Chemiluminescent Substrate Reagent Kit (Thermo Fisher) in the ChemiDoc™ Imaging System (Bio-Rad).

## *In vivo* assays

The IDIBELL's Animal Care and Use Committee approved the animal studies. Six-week-old female C57BL/6J mice were purchased from Charles River, inoculated with 105 K cells ($2.5 \times 10^6$ cells) in Matrigel (Corning). When tumors attained a volume of 50–100 mm³ (monotherapy) or 150–200 mm³ (rapamycin and its combinations), they were randomly assigned to control and treatment groups. To reduce variability, a single researcher measured tumor volume through all the 105 K-based assays. The applied doses of drugs or compounds were based on those used in previous animal studies

(Haberle *et al*, 2002; Francis *et al*, 2012; Ledesma *et al*, 2014; Cheng *et al*, 2016; Chen *et al*, 2017). The drugs were administrated as follows: vehicle control (0.2% carboxymethylcellulose (CMC) and 0.25% Tween-80; intraperitoneal), rapamycin (1 or 0.25 mg/kg/day in CMC; intraperitoneal), clorgyline (10 mg/kg/day in saline; intraperitoneal), loratadine (25 mg/kg/day in CMC; oral), and rasagiline (1 mg/kg/day in saline; intraperitoneal). The α-methyl-DL-histidine was administrated 150 mg/kg/day in saline, intraperitoneal. Treatment regimes involved consecutive intervals of 5 days of drug or compound administration, and 2 days of rest; except for loratadine, which was administered every other day for 3 days a week. The rapamycin dose of 0.25 mg/kg/day has been previously tested in EL4 lymphoma cells where it reduced phosphor-ribosomal protein S6 signaling and altered immune cell content in tumor-draining lymph nodes (Liu *et al*, 2017). In the combined assays of rapamycin with another drug, the mTOR inhibitor was administered first followed by the other drug approximately 1 h later. Before the formal assays, the following drugs and concentrations were tested for potential toxicities or effects: rasagiline 0.1 and 1 mg/kg/day; and rapamycin 0.05, 0.1, 0.25, 0.5, and 1 mg/kg/day. The 0.25 mg/kg/day dose was the lowest that stabilized the tumors. For shRNA-based assays, the following constructs were chosen and validated from among those available from the MISSION library (Sigma-Aldrich): shMaoa TRCN0000327502 and TRCN0000327503; and shHrh1 TRCN0000028707. The 105K cells (2.5 million) were injected subcutaneously and tumor growth was measured weekly, starting at a volume of approximately 50 mm³. For the L-histidine diet assays, the mice were provided the specific diet 48 h before 105 K cell injections, and the diet was maintained throughout the experiment while measuring tumor growth; the mouse meals were checked every working day. Mice were housed and maintained in laminar flow cabinets under specific pathogen-free conditions. All the animal studies were approved by the local committee for animal care (IDIBELL, protocol #10451).

## Data availability

The RNA-seq data of *Tsc2*-deficient 105 K treated tumors have been deposited under the Gene Expression Omnibus reference GSE173332 (https://www.ncbi.nlm.nih.gov/geo/query/acc.cgi?acc = GSE173332).

**Expanded View** for this article is available online.

## Acknowledgements

We dedicate this work to the memory of Dr. Antoni Xaubet, who helped initiate LAM studies. We thank the individuals who provided their blood samples, and the AELAM foundation for its continued support for LAM research. We also thank Prof. E. Henske for providing the LAM cell models. The metabolic studies were performed thanks to the technological infrastructure of the Centre for Omic Sciences (COS), Nutrition and Health Technology Centre (CTNS), Reus, Spain. This research was supported by AELAM, The LAM Foundation (Seed Grant 2019), Instituto de Salud Carlos III grants PI15/00854, PI18/01029, and ICI19/00047 (co-funded by European Regional Development Fund (ERDF), a way to build Europe), Generalitat de Catalunya SGR grants 2014-364 and 2017-449, the CERCA Program, and ZonMW-TopZorg grant 842002003. C.L.M. acknowledges the financial support (PRA-2017-51 project) of the University of Pisa. A.U.K. is supported by Nottingham Trent University's Independent

Fellowship Scheme. Our results are partly based upon data generated by the TCGA Research Network (https://www.cancer.gov/tcga), and we express our gratitude to the TCGA consortia and coordinators for producing the data and the clinical information used in our study.

## Author contributions

MAP designed the study and wrote the paper; CH, FM, and NG performed the *in vivo* experiments; CH, FM, NG, AIE, JMC, RG, CG, and MAP analyzed the *in vivo* experimental data; CH, FM, GRG, and AB performed the *in vitro* experiments; CH, FM, GRG, AB, JAM, JCP, FV, AA, XZ, HF, RH, CG, MF, and MAP analyzed the *in vitro* experimental data; AJ, JC, and OY performed the metabolite analyses; AG, LP, RE, EB, and MAP performed the bioinformatic analyses; AV and CB evaluated the pathology samples; CH, FM, AG, LP, DC, and MAP did the statistical analyses; SRJ and SM performed the VEGF-D determinations; SRJ, SM, ER-L, BS, SG-O, JA, CV, TA, PU, RL, AX, JAR-P, AM-W CM, JB, EL, CHMM, JJV, MJRQ, AC, MM-M, and AR collected patient samples and clinical data; CLM produced GA11; AN-C and MM provided histamine receptors data; RR-B performed rapamycin quantifications; CH, AU-K, and EB performed the MAO activity assays, analyzed, and interpreted the data; CH, FM, SRJ, SZ, MM-M, AR, and OY evaluated the output for its significant intellectual content.

## Conflict of interest

M.A.P. is recipient of an unrestricted research grant from Roche Pharma for the development of the ProCURE ICO research program.

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
