## [Review Process File · EMBO Molecular Medicine]

Histamine signaling and metabolism identify potential biomarkers and therapies for lymphangiomyomatosis

Carmen Herranz, Francesca Mateo, Alexandra Baiges, Gorka Ruiz de Garibay, Alexandra Junza, Simon Johnson, Suzanne Miller, Nadia García, Jordi Capellades, Antonio Gomez, August Vidal, Luis Palomero, Roderic Espín, Ana Extremera, Eline Blommaert, Eva Revilla-López, Berta Saez, Susana Gómez-Ollés, Julio Ancochea, Claudia Valenzuela, Tamara Alonso, Piedad Ussetti, Rosalía Laporta, Antoni Xaubet, José Rodríguez-Portal, Ana Montes-Worboys, Carlos Machahua, Jaume Bordas-Martinez, Javier Menedez, Josep Cruzado, Roser Guiteras, Christophe Bontoux, Concettina La Motta, Aleix Noguera-Castells, Mario Mancino, Enrique Lastra, Raúl Rigo-Bonnin, José Carlos Perales, Francesc Viñals, Alvaro Lahiguera, Xiaohu Zhang, Daniel Cuadras, Coline van Moorsel, Joanne van der Vis, Marian Quanjel, Harilaos Filippakis, Razq Hakem, Chiara Gorrini, Marc ferrer, Aslihan Ugun-Klusek, Elizabeth Billett, Elżbieta Radzikowska, Álvaro Casanova, María Molina-Molina, Antonio Román, Oscar Yanes, and Miquel Angel Pujana

DOI: [10.15252/emmm.202113929](https://doi.org/10.15252/emmm.202113929)

Corresponding author: Miquel Angel Pujana (miguelangel.pujana@gmail.com)

Review Timeline:

Submission Date:	9th Jan 21
Editorial Decision:	22nd Feb 21
Revision Received:	5th May 21
Editorial Decision:	15th Jun 21
Authors' Correspondence:	9th July 21
Editor's Correspondence:	15th July 21
Revision Received:	19th Jul 21
Accepted:	21st Jul 21

Editor: Lise Roth

Transaction Report:

22nd Feb 2021

Dear Dr. Pujana,

Thank you for the submission of your manuscript to EMBO Molecular Medicine, and please accept my apologies for the unusual delay in getting back to you. As mentioned in my previous correspondence, two out of the three referees who initially agreed to review your manuscript failed to provide their reports in a timely manner, and I therefore contacted additional referees.

We have now received feedback from three of the four reviewers who eventually agreed to evaluate your manuscript. Referee #4 has not provided his/her report yet, but given that the other referees are overall positive, we prefer to make a decision now in order to avoid further delay in the process. Should referee #4 provide a report, we will send it to you, with the understanding that we would not ask you for extensive experiments in addition to the ones required in the enclosed reports.

As you will see from the reports below, the referees acknowledge the interest of the study and are overall supporting publication of your work pending appropriate revisions.

Addressing the reviewers' concerns in full will be necessary for further considering the manuscript in our journal, and acceptance of the manuscript will entail a second round of review. EMBO Molecular Medicine encourages a single round of revision only and therefore, acceptance or rejection of the manuscript will depend on the completeness of your responses included in the next, final version of the manuscript. For this reason, and to save you from any frustrations in the end, I would strongly advise against returning an incomplete revision.

When submitting your revised manuscript, please carefully review the instructions that follow below. Failure to include requested items will delay the evaluation of your revision:

- 1) A .docx formatted version of the manuscript text (including legends for main figures, EV figures and tables). Please make sure that the changes are highlighted to be clearly visible.
- 2) Individual production quality figure files as .eps, .tif, .jpg (one file per figure).
- 3) A .docx formatted letter INCLUDING the reviewers' reports and your detailed point-by-point responses to their comments. As part of the EMBO Press transparent editorial process, the point-by-point response is part of the Review Process File (RPF), which will be published alongside your paper.
- 4) A complete author checklist, which you can download from our author guidelines (<https://www.embopress.org/page/journal/17574684/authorguide#submissionofrevisions>). Please insert information in the checklist that is also reflected in the manuscript. The completed author checklist will also be part of the RPF.
- 5) Before submitting your revision, primary datasets produced in this study need to be deposited in

an appropriate public database (see <https://www.embopress.org/page/journal/17574684/authorguide#dataavailability>). Please remember to provide a reviewer password if the datasets are not yet public. The accession numbers and database should be listed in a formal "Data Availability" section (placed after Materials & Method). Please note that the Data Availability Section is restricted to new primary data that are part of this study.

"This study includes no data deposited in external repositories"

6) We would also encourage you to include the source data for figure panels that show essential data. Numerical data should be provided as individual .xls or .csv files (including a tab describing the data). For blots or microscopy, uncropped images should be submitted (using a zip archive if multiple images need to be supplied for one panel). Additional information on source data and instruction on how to label the files are available at .

7) Our journal encourages inclusion of *data citations in the reference list* to directly cite datasets that were re-used and obtained from public databases. Data citations in the article text are distinct from normal bibliographical citations and should directly link to the database records from which the data can be accessed. In the main text, data citations are formatted as follows: "Data ref: Smith et al, 2001" or "Data ref: NCBI Sequence Read Archive PRJNA342805, 2017". In the Reference list, data citations must be labeled with "[DATASET]". A data reference must provide the database name, accession number/identifiers and a resolvable link to the landing page from which the data can be accessed at the end of the reference. Further instructions are available at .

8) We replaced Supplementary Information with Expanded View (EV) Figures and Tables that are collapsible/expandable online. A maximum of 5 EV Figures can be typeset. EV Figures should be cited as 'Figure EV1, Figure EV2" etc... in the text and their respective legends should be included in the main text after the legends of regular figures.

- Additional Tables/Datasets should be labeled and referred to as Table EV1, Dataset EV1, etc. Legends have to be provided in a separate tab in case of .xls files. Alternatively, the legend can be supplied as a separate text file (README) and zipped together with the Table/Dataset file. See detailed instructions here: .

9) The paper explained: EMBO Molecular Medicine articles are accompanied by a summary of the articles to emphasize the major findings in the paper and their medical implications for the non-specialist reader. Please provide a draft summary of your article highlighting

10) For more information: There is space at the end of each article to list relevant web links for further consultation by our readers. Could you identify some relevant ones and provide such information as well? Some examples are patient associations, relevant databases, OMIM/proteins/genes links, author's websites, etc...

11) Every published paper now includes a 'Synopsis' to further enhance discoverability. Synopses are displayed on the journal webpage and are freely accessible to all readers. They include a short stand first (maximum of 300 characters, including space) as well as 2-5 one-sentences bullet points that summarizes the paper. Please write the bullet points to summarize the key NEW findings. They should be designed to be complementary to the abstract - i.e. not repeat the same text. We encourage inclusion of key acronyms and quantitative information (maximum of 30 words / bullet point). Please use the passive voice. Please attach these in a separate file or send them by email, we will incorporate them accordingly.

Please also suggest a striking image or visual abstract to illustrate your article. If you do please provide a png file 550 px-wide x 400-px high.

12) As part of the EMBO Publications transparent editorial process initiative (see our Editorial at <http://embomolmed.embopress.org/content/2/9/329>), EMBO Molecular Medicine will publish online a Review Process File (RPF) to accompany accepted manuscripts.

In the event of acceptance, this file will be published in conjunction with your paper and will include the anonymous referee reports, your point-by-point response and all pertinent correspondence relating to the manuscript. Let us know whether you agree with the publication of the RPF and as here, if you want to remove or not any figures from it prior to publication.

I look forward to receiving your revised manuscript.

Yours sincerely,

Lise Roth

Lise Roth, PhD
Editor
EMBO Molecular Medicine

***** Reviewer's comments *****

Referee #1 (Remarks for Author):

The manuscript by Dr. Pujana and colleagues report novel observations about identifying MIAA as a novel biomarker for patients with rare disease pulmonary Lymphangioliomyomatosis (LAM). Extensive and strong experimental data demonstrate dysregulation of histamine signaling and metabolism as it relates to TSC2 loss, mTORC1 upregulation and LAM disease. Study also explore repurposing loratadine for potential treatment of patients with LAM. Overall this is very important, strong and well executed and written study, with potential to change clinical practice and impact how LAM is treated in near future. I have a minor suggestion to include two latest references about mitochondrial dysfunction affecting mTOR1 and data from another study of scRNA-seq of LAM2.

1. Condon, K. J. et al. Genome-wide CRISPR screens reveal multitiered mechanisms through which mTORC1 senses mitochondrial dysfunction. *Proc. Natl. Acad. Sci.* 118, e2022120118 (2021).
2. Obraztsova, K. et al. mTORC1 activation in lung mesenchyme drives sex- and age-dependent pulmonary structure and function decline. *Nat. Commun.* 11, 5640 (2020).

Referee #2 (Comments on Novelty/Model System for Author):

This article demonstrates the role of Histamine signaling and metabolism in the identification of biomarkers and the therapeutic opportunities for lymphangioliomyomatosis (LAM). They also showed the role of Monoamine oxidase A/B and Histamine H1 receptor in this process. They categorically showed that targeting monoamine oxidase A/B by clorgyline and rasagiline or by targeting Histamine H1 receptor loratidine the LAM tumorigenesis can be reduced. They further reveal that these drugs combined with rapamycin (the mTOR inhibitor) can produce synergistic effect. This study is novel, technically sound and highly interesting. This study further addresses a very relevant issue which has high biomedical implication. Moreover the study was carried out in cellular model, in vivo system and with patients tissue samples. I appreciate the effort by the group and in favour of accepting this manuscript after minor revision.

Referee #2 (Remarks for Author):

The authors show a very interesting study in this manuscript by highlighting the role of histamine metabolites as biomarkers and the therapeutic opportunities in LAM management. The data are solid and well done. In most of the cases, the study was conducted with the pharmacological inhibitors that should be confirmed by using shRNA of MAO A/B or by shHrh1 (like in case of Fig.7A,B). The combinatorial effect in presence of Rapamycin should be done in presence of those siRNAs.

The crucial point is how MAO A/B activation and HRH1 metabolism and signaling is related to mTOR pathway? Is there any role of PI3K/AKT in this process?

As they claimed that shRNAs for MAOA/Hrh1 showed epitheloid morphology , it raises the question

whether they are inducing the expression of epithelial markers and reducing mesenchymal markers in LAM cellular model?

Whether MAO-A activation and corresponding histamine-derived metabolites are linked to migration/invasion in LAM cell model? These things need to be answered before the acceptance of this manuscript.

Referee #3 (Comments on Novelty/Model System for Author):

Technically this is a sound study. A concern over statistical methods of analysis is explained in the comments to the authors (minor concern #2).

This is a novel study.

Based on the presented data and the small number of LAM patients, the medical impact of this study would not be high.

The model systems (cell lines, model organisms, methodologies) are appropriate.

Referee #3 (Remarks for Author):

This work implicates histamine signaling in the pathology of the rare pulmonary disease LAM.

The authors perform a comprehensive analysis of histamine signaling metabolites, primarily methylimidazoleacetic acid (MIAA) and demonstrate that although MIAA does not correlate well with VEGF-D plasma levels, when used in combination with VEGF-D it can increase the predictive value in LAM patients vs. patients from other pulmonary diseases (namely Langerhans, Sjorgen, Lupus and Emphysema).

Next, the authors demonstrate that genes in the histamine signaling pathway (ALDH2, MAOA and MAOB) have higher expression at the mRNA and protein levels in Tsc2-null cells and LAM tumors, and they implicate some of these genes in metabolism and increased ROS and mt oxidation in Tsc2-null cells.

They provide evidence that the HRH1 antagonist loratadine synergizes with mTORC1 inhibitors to decrease Tsc2-null cell viability in vitro, although there is not any evidence of apoptosis. In 105k isograft mice, loratadine (HRH1 antagonist), clorgyline (MAO-A inhibitor) and rasagiline (MAO-B inhibitor) all inhibited significantly tumor growth, but failed to eliminate the tumors. All three drugs when combined singly with rapamycin provided better tumor growth inhibition, compared to rapamycin alone. Similar to the in vitro studies, there is no evidence of apoptosis in the 105k isograft tumors treated with the loratadine, clorgyline and rasagiline - either singly or in combination with rapamycin.

Overall, this is a strong study resulting in novel hypotheses regarding the pathogenesis of LAM disease and proposing new drugable targets with clinical significance. The conclusions are drawn based on strong evidence and are not overstated. Addressing the following concerns would certainly enhance the quality, significance and visibility of this work.

Minor concerns

1. Based on the marginal increase of MIAA blood levels in LAM vs. controls, I am skeptical for the use

of the term "biomarker, especially in the main title of the article.

2. A Mann-Whitney test for metabolite analyses between multiple groups (Figure 1, several panels) may be an inadequate statistical method due to the variability of these samples. A statistician with relevant experience should be consulted whether a one-way ANOVA would be more appropriate.
3. When determining synergy between loratadine and everolimus (Fig. 5F), why a non-constant ratio method was chosen vs. a more standard constant ratio method? (see ref. with PMID: 20068163).

Major concerns

1. *** When studying the effects of drug combinations on the inhibition of 105k tumor growth (Fig. 6C and 6D) the authors need to take into consideration that both rapamycin and loratadine are metabolized by the same enzyme (p450 3A4, CYP3A4). Commercial drug databases report significant interaction between these 2 drugs, such to warrant close monitoring of drug levels. Could the effects of the loratadine+rapamycin combination on tumor size and mass be due to increases in rapamycin trough levels? Are the rapamycin trough levels changed in mice receiving combination vs. those with rapamycin monotherapy? This should be addressed by examining drug levels in blood specimens from treated mice, or by conducting PK studies for the drug doses used. This question becomes even more critical since there is a study conducted in LAM patients receiving loratadine+rapamycin (vs. rapamycin alone) and it would be very important to monitor drug trough levels in these patient cohorts.

Herranz *et al.*, Histamine signaling and metabolism identify potential biomarkers and therapies for lymphangioleiomyomatosis" (manuscript EMM-2021-13929).

Reviewer #1

The manuscript by Dr. Pujana and colleagues report novel observations about identifying MIAA as a novel biomarker for patients with rare disease pulmonary Lymphangioleiomyomatosis (LAM). Extensive and strong experimental data demonstrate dysregulation of histamine signaling and metabolism as it relates to TSC2 loss, mTORC1 upregulation and LAM disease. Study also explore repurposing loratadine for potential treatment of patients with LAM. Overall this is very important, strong and well executed and written study, with potential to change clinical practice and impact how LAM is treated in near future. I have a minor suggestion to include two latest references about mitochondrial dysfunction affecting mTOR1 and data from another study of scRNA-seq of LAM2.

1. Condon, K. J. et al. Genome-wide CRISPR screens reveal multitiered mechanisms through which mTORC1 senses mitochondrial dysfunction. *Proc. Natl. Acad. Sci.* 118, e2022120118 (2021).
2. Obraztsova, K. et al. mTORC1 activation in lung mesenchyme drives sex- and age-dependent pulmonary structure and function decline. *Nat. Commun.* 11, 5640 (2020).

We are very grateful for the comments about the quality and relevance of our work. We acknowledge the relevance of recent studies noted by the reviewer and we have cited them in the corresponding sentence (page 12, lines 202-204).

Reviewer #2

This article demonstrates the role of Histamine signaling and metabolism in the identification of biomarkers and the therapeutic opportunities for lymphangioliomyomatosis (LAM). They also showed the role of Monoamine oxidase A/B and Histamine H1 receptor in this process. They categorically showed that targeting monoamine oxidase A/B by clorgyline and rasagiline or by targeting Histamine H1 receptor loratidine the LAM tumorigenesis can be reduced. They further reveal that these drugs combined with rapamycin (the mTOR inhibitor) can produce synergistic effect. This study is novel, technically sound and highly interesting. This study further addresses a very relevant issue which has high biomedical implication. Moreover the study was carried out in cellular model, in vivo system and with patients tissue samples. I appreciate the effort by the group and in favour of accepting this manuscript after minor revision.

We are very grateful for the comments about the novelty, quality, and interest of our study.

The authors show a very interesting study in this manuscript by highlighting the role of histamine metabolites as biomarkers and the therapeutic opportunities in LAM management. The data are solid and well done. In most of the cases, the study was conducted with the pharmacological inhibitors that should be confirmed by using shRNA of MAO A/B or by shHrh1 (like in case of Fig.7A,B). The combinatorial effect in presence of Rapamycin should be done in presence of those siRNAs.

We agree that the suggested assays would have further supported the inhibitory combination effect of rapamycin and MAO/HRH1-targeting drugs. To address this, we performed additional *in vitro* and *in vivo* assays. *In vitro*, we measured proliferation of *Tsc2*-deficient 105K cells exposed to rapamycin (20 nM) and/or transduced with shRNAs against *Hrh1* or *Maoa* expression (these assays were carried out on two separate occasions, with six replicates for each condition/time point). Confirming the original *in vivo* observations, single shRNA transductions significantly reduced cell proliferation relative to pLKO-DMSO treated cells, and the combination of rapamycin with shRNA-*Maoa* reduced proliferation significantly more than rapamycin alone (two-tailed t-test $P < 0.05$; **Appendix Fig S10A**, page 17, lines 318-322). The combination with shRNA-*Hrh1* did not show similar effects *in vitro*. We also tested the rapamycin-shRNAs combinations *in vivo* by measuring tumor weight differences at the end of treatments: pLKO, shRNA-*Hrh1*, shRNA-*Maoa* and/or treated with rapamycin (0.25 mg/kg/day) or vehicle. Tumor growth was further reduced when *Tsc2*-deficient 105K cells were transduced with shRNA-*Hrh1* or shRNA-*Maoa* and also treated with rapamycin, relative to cells transduced with pLKO and similarly treated with rapamycin (two-tailed Mann-Whitney $P = 0.013$ and $P = 0.0009$, respectively; **Appendix Fig S10B**, page 17, lines 322-325). These additional data confirm the original observations based on drug combinations tested *in vivo*. We have also completed the study of the original drug assays by quantifying Ki67 staining in the corresponding tumors (**Appendix Fig S15**).

The crucial point is how MAO A/B activation and HRH1 metabolism and signaling is related to mTOR pathway? Is there any role of PI3K/AKT in this process?

We agree that the connection between histamine metabolism/signaling and mTOR pathway signaling emerge as an intriguing and relevant point in our study. Interestingly, we noted that cancer cell lines responses to sirolimus and loratadine in the NCI-60 dataset were positively correlated independently of tumor type (**Appendix Fig S11**, page 19, lines 353-355). Then, we analyzed the levels of phosphorylation/total AKT and ribosomal S6 protein in *Tsc2*-deficient MEF and 105K cells exposed *in vitro* to vehicle-DMSO, each monotherapy, and rapamycin combinations. The monotherapies of clorgyline, loratadine, and rasagiline showed relative reductions of phospho-Ser235/236 and total S6, although to a lesser extent than rapamycin monotherapy (**Appendix Fig S12**, page 19, lines 355-361). We speculate that histamine signaling could partially converge on regulation of S6 activity/levels, possibly as indicated in other cellular types (PMID: 11959800). However, we also acknowledge that further studies are warranted to precisely determine the underlying signaling cross-talk mechanism (**Discussion**, page 23, lines 431-433).

As they claimed that shRNAs for MAOA/Hrh1 showed epitheloid morphology, it raises the question whether they are inducing the expression of epithelial markers and reducing mesenchymal markers in LAM cellular model?

The reviewer raises another accurate point. To address this, we took complementary approaches. As evidence from loratadine has led to a clinical trial, we performed RNA-seq analyses of the original *Tsc2*-deficient 105K tumors treated with vehicle, rapamycin, loratadine, or a combination of rapamycin and loratadine (two tumors for each group), and assessed differentially represented GO terms relative to vehicle. The loratadine setting was the one with most gene expression changes linked to cell differentiation and development (**Appendix Fig S16**, page 19, lines 370-372). The RNA-seq data have been deposited under GEO accession number GSE173332. Then, to specifically assess diseased cell changes, we transduced *Tsc2*-deficient 105K cells *in vitro* with the shRNAs against *Maoa* and *Hrh1*, and quantified the expression of informative markers relative to pLKO control (three independent experiments, three replicates per condition). The expression of *Epcam* was found to be increased, while of mesenchymal markers (*Fn1*, *Snai1*, *Vim*, and *Twist1*) decreased with depletion of *Maoa* or *Hrh1* (**Appendix Fig S19**, page 20, lines 377-382).

Whether MAO-A activation and corresponding histamine-derived metabolites are linked to migration/invasion in LAM cell model? These things need to be answered before the acceptance of this manuscript.

To address this question, we performed transwell migration and wound healing assays using *Tsc2*-deficient 105K cells (two independent experiments of each assay type, and three replicates in each condition). In transwell migration assays, shRNA-mediated depletion of *Maoa* or *Hrh1* expression caused a significant reduction in the number of invasive cells (**Appendix Fig S20A**). Similarly, depletion of *Maoa* or *Hrh1* expression caused a significant reduction in wound closure (**Appendix Fig S20B**, page 20, lines 382-385). The *in vitro* effects appeared to be relatively stronger with depletion of *Hrh1* expression.

Referee #3

Technically this is a sound study. A concern over statistical methods of analysis is explained in the comments to the authors (minor concern #2). This is a novel study. Based on the presented data and the small number of LAM patients, the medical impact of this study would not be high. The model systems (cell lines, model organisms, methodologies) are appropriate.

This work implicates histamine signaling in the pathology of the rare pulmonary disease LAM. The authors perform a comprehensive analysis of histamine signaling metabolites, primarily methylimidazoleacetic acid (MIAA) and demonstrate that although MIAA does not correlate well with VEGF-D plasma levels, when used in combination with VEGF-D it can increase the predictive value in LAM patients vs. patients from other pulmonary diseases (namely Langerhans, Sjorgen, Lupus and Emphysema).

Next, the authors demonstrate that genes in the histamine signaling pathway (ALDH2, MAOA and MAOB) have higher expression at the mRNA and protein levels in Tsc2-null cells and LAM tumors, and they implicate some of these genes in metabolism and increased ROS and mt oxidation in Tsc2-null cells.

They provide evidence that the HRH1 antagonist loratadine synergizes with mTORC1 inhibitors to decrease Tsc2-null cell viability in vitro, although there is not any evidence of apoptosis. In 105k isograft mice, loratadine (HRH1 antagonist), clorgyline (MAO-A inhibitor) and rasagiline (MAO-B inhibitor) all inhibited significantly tumor growth, but failed to eliminate the tumors. All three drugs when combined singly with rapamycin provided better tumor growth inhibition, compared to rapamycin alone. Similar to the in vitro studies, there is no evidence of apoptosis in the 105k isograft tumors treated with the loratadine, clorgyline and rasagiline - either singly or in combination with rapamycin.

Overall, this is a strong study resulting in novel hypotheses regarding the pathogenesis of LAM disease and proposing new drugable targets with clinical significance. The conclusions are drawn based on strong evidence and are not overstated. Addressing the following concerns would certainly enhance the quality, significance and visibility of this work.

We appreciate the comments on the robustness, quality, and interest of our results. Also, we acknowledge the following points, which we have accommodated to improve the presentation and discussion of our study.

Minor concerns

1. Based on the marginal increase of MIAA blood levels in LAM vs. controls, I am skeptical for the use of the term "biomarker, especially in the mail title of the article.

The evidence for MIAA as potential biomarker was derived from plasma-based quantifications in two patient cohorts, by metabolic profiling of LAM cell models, and confirmed by analyzing plasma of mice carrying *Tsc2*-deficient 105K tumors. The evidence is also supported by quantification of histamine in the replication cohort. However, we acknowledge that the patient studies were retrospective, and that the

marker differences between controls and cases were not sufficiently decisive to permit categorization, so we have therefore edited the title, abstract, and introduction (final paragraph) to include the notion of “potential”.

2. A Mann-Whitney test for metabolite analyses between multiple groups (Figure 1, several panels) may be an inadequate statistical method due to the variability of these samples. A statistician with relevant experience should be consulted whether a one-way ANOVA would be more appropriate.

We respectfully consider that the Mann-Whitney test was properly applied for comparing pairs of groups given that the data were not normally distributed. Preceded by a discovery LAM sample set, we validated the LAM-centered differences by applying the Mann-Whitney test for comparisons between LAM and each other control/disease group, or between specific conditions (i.e., treatment with rapamycin yes/no). However, the reviewer is correct this test cannot be used to compare three or more independent groups. The equivalent nonparametric test of the ANOVA (which cannot be used because the data are not normally distributed; Shapiro-Wilk test) is the Kruskal-Wallis test. For this reason, we also note the results of these tests noted in the legends of panels 1C and 1G.

3. When determining synergy between loratadine and everolimus (Fig. 5F), why a non-constant ratio method was chosen vs. a more standard constant ratio method? (see ref. with PMID: 20068163).

The reviewer raises another salient point. We did indeed assess different concentrations of the rapalog *in vitro* and observed no substantial differences across a range of about 100-fold; therefore we decided to use a constant concentration in subsequent assays. Please note below the results of a typical study with a range of concentrations of everolimus in *Tsc2*-wild-type and *Tsc2*-deficient MEF cultures.

Major concerns

1. * When studying the effects of drug combinations on the inhibition of 105k tumor growth (Fig. 6C and 6D) the authors need to take into consideration that both rapamycin and loratadine are metabolized by the same enzyme (p450 3A4, CYP3A4). Commercial drug databases report significant interaction between these 2 drugs, such to warrant close monitoring of drug levels. Could the effects of the loratadine+rapamycin combination on tumor size and mass be due to increases in rapamycin trough levels? Are the rapamycin trough levels changed in mice**

receiving combination vs. those with rapamycin monotherapy? This should be addressed by examining drug levels in blood specimens from treated mice, or by conducting PK studies for the drug doses used. This question becomes even more critical since there is a study conducted in LAM patients receiving loratadine+rapamycin (vs. rapamycin alone) and it would be very important to monitor drug trough levels in these patient cohorts.

We agree that this was a key missing point in our original submission and we thank the reviewer for noting its relevance relative to the upcoming trial. To address this issue, we engrafted *Tsc2*-deficient 105K cells in C57BL/6J mice and, when tumor volume reached 100-150 mm³, treated the animals as originally reported with rapamycin or rapamycin plus loratadine. Using a Certified Clinical Laboratory facility (additional authorship in this submission; R Rigo-Bonnin, head of the laboratory that routinely measures rapamycin for patient treatment in our general hospital) we quantified rapamycin in mouse plasma using a time course approach, after 24 hours of drug administration, and in tumors at the end of the study and after 24 hours of drug administration. In none of these three tests, the levels of rapamycin were found to be significantly different between the rapamycin monotherapy group and the rapamycin-loratadine combination (**Appendix Fig S8**, page 17, lines 309-313).

15th Jun 2021

Dear Dr. Pujana,

Thank you for the submission of your revised manuscript to EMBO Molecular Medicine, and please accept my apologies for the delay in getting back to you, as we were expecting two additional referees' reports. While one report is still missing, referees #1 and #2 kindly reviewed your answers to all referees and were satisfied. I am therefore pleased to inform you that we will be able to accept your manuscript once the following editorial points will be addressed:

1/ Main manuscript text:

- Please answer/correct the changes suggested by our data editors in the main manuscript file (in track changes mode). This file will be sent in the next couple of days. Please use this file for any further modification.
- Please provide up to 5 keywords.
- Material and methods:
 - o Patients and samples: please include a statement that the experiments conformed to the principles set out in the WMA Declaration of Helsinki and the Department of Health and Human Services Belmont Report.
 - o Cell lines and media: please indicate whether the cells were tested for mycoplasma contamination and authenticated.
 - o In vivo assays: please indicate the housing and husbandry conditions of the mice
- Thank you for providing a Data Availability section. Please provide a direct link to access the data and note that the data have to be publicly available before acceptance of the manuscript.
- Authors' contribution: Jaume Bordas, Joanne van der Vis, Marian Quanjel, Chiara Gorrini are missing. Please also clarify which is the correct spelling: Charilaos Filippakis (submission system) or Harilaos Filippakis (manuscript).
- Please replace "Competing interests" by "Conflict of interest".
- Funding: AELAM, CERCA Program, Nottingham Trent University's Independent Fellowship Scheme are not entered in the submission system, please update.

2/ Figures and Appendix

- Please indicate in the main and appendix figures or in their legends the exact n= and exact p= values, not a range, along with the statistical test used, including for non-significant p-values. Some people found that to keep the figures clear, providing a supplemental table in the appendix with all exact p-values was preferable. You are welcome to do this if you want to.
- Appendix: please merge all appendix figures and tables into 1 pdf and add a table of content. Please update the callout to "Appendix Figure S1" etc and "Appendix Table S1" etc. Please remove the appendix figure legends from the main manuscript file and add them to the appendix file (below each figure).
- Appendix Fig S20: please add and define scale bars.

3/ We would also encourage you to include the source data for figure panels that show essential data. Numerical data should be provided as individual .xls or .csv files (including a tab describing the data). For blots or microscopy, uncropped images should be submitted (using a zip archive if multiple images need to be supplied for one panel). Additional information on source data and instruction on how to label the files are available at

4/ The paper explained: EMBO Molecular Medicine articles are accompanied by a summary of the articles to emphasize the major findings in the paper and their medical implications for the non-specialist reader. Please provide a draft summary of your article highlighting:

5/ Every published paper includes a 'Synopsis' to further enhance discoverability. Synopses are displayed on the journal webpage and are freely accessible to all readers. They include a short stand first (maximum of 300 characters, including space) as well as 2-5 one-sentences bullet points that summarizes the paper. Please write the bullet points to summarize the key NEW findings. They should be designed to be complementary to the abstract - i.e. not repeat the same text. We encourage inclusion of key acronyms and quantitative information (maximum of 30 words / bullet point). Please use the passive voice. Please attach these in a separate file or send them by email, we will incorporate them accordingly.

6/ As part of the EMBO Publications transparent editorial process initiative (see our Editorial at <http://embomolmed.embopress.org/content/2/9/329>), EMBO Molecular Medicine will publish online a Review Process File (RPF) to accompany accepted manuscripts.

This file will be published in conjunction with your paper and will include the anonymous referee reports, your point-by-point response and all pertinent correspondence relating to the manuscript. Let us know whether you agree with the publication of the RPF and as here, if you want to remove or not any figures from it prior to publication.

I look forward to receiving your revised manuscript.

Yours sincerely,

Lise Roth

Lise Roth, PhD
Editor
EMBO Molecular Medicine

***** Reviewer's comments *****

Referee #1 (Remarks for Author):

suitable for publication.

Referee #2 (Remarks for Author):

Thank you so much to address all the concerns raised by me so elaborately. I am totally convinced by your responses to the revised version of the manuscript.

When analyzing the original data corresponding to tumor weights shown in Figure 6D, by mistake I mixed sample identifiers in the code and the results were not completely correct in our initial submission.

Although similar trends can be observed, the comparisons between control and clorgyline, and between rapamycin and rapamycin plus loratadine do not reach significance.

I regret very much this error and we are willing to follow your indications. If considered, we are willing to repeat the corresponding in vivo assays. Again, I am very sorry for this issue, but wanted to be honest on exposing it.

Thank you very much for submitting your revised files. Regarding the new data in Fig 6, referee #1 agreed that the trend is not changed compared to the previous version, and that these new data do not alter the conclusions of the manuscript. No additional experiment will therefore be needed.

The authors performed the requested editorial changes.

21st Jul 2021

Dear Miquel Angel,

Thank you for providing the revised files and bearing with the last editorial changes. I am pleased to inform you that your manuscript is accepted for publication and is now being sent to our publisher to be included in the next available issue of EMBO Molecular Medicine!

Please read below for additional IMPORTANT information regarding your article, its publication and the production process.

Congratulations on your interesting work!

With my best wishes,

Lise

Lise Roth, Ph.D
Editor
EMBO Molecular Medicine

Follow us on Twitter @EmboMolMed
Sign up for eTOCs at embopress.org/alertsfeeds

Corresponding Author Name: MIQUEL ANGEL PUJANA

Manuscript Number: EMM-2021-13929